# Identifying the geometric catalytic active sites of crystalline cobalt oxyhydroxides for oxygen evolution reaction

Sihong Wang[1], Qu Jiang [1], Shenghong Ju[2], Chia-Shuo Hsu [3], Hao Ming Chen [3,4], Di Zhang [1] & Fang Song [1] ✉

Unraveling the precise location and nature of active sites is of paramount significance for the understanding of the catalytic mechanism and the rational design of efficient electrocatalysts. Here, we use well-defined crystalline cobalt oxyhydroxides CoOOH nanorods and nanosheets as model catalysts to investigate the geometric catalytic active sites. The morphology-dependent analysis reveals a ~50 times higher specific activity of CoOOH nanorods than that of CoOOH nanosheets. Furthermore, we disclose a linear correlation of catalytic activities with their lateral surface areas, suggesting that the active sites are exclusively located at lateral facets rather than basal facets. Theoretical calculations show that the coordinatively unsaturated cobalt sites of lateral facets upshift the O 2p-band center closer to the Fermi level, thereby enhancing the covalency of Co-O bonds to yield the reactivity. This work elucidates the geometrical catalytic active sites and enlightens the design strategy of surface engineering for efficient OER catalysts.

Water splitting into hydrogen and oxygen molecules is a promising solution for sustainable and eco-friendly energy storage from renewable power sources[1–4]. Oxygen evolution reaction (OER) is the rate-determining process and contributes a substantial overpotential and energy loss to the overall water splitting[5–9]. Numerous transition metal oxides have been explored to lower the overpotential, among which cobalt-based materials have been regarded as strong candidates[10–16]. Unfortunately, the catalytic activity is moderate, requiring overpotentials of more than 300 mV to deliver a current density of 10 mA cm$^{-2}$ on flat electrodes.

Unraveling the precise location of active sites has highly fundamental and technological implications for the rational design of next-generation electrocatalysts[12,15,17–21]. Due to the heterogeneous nature of electrocatalysts and an extensive library of crystal structures (such as (oxy)hydroxide, spinel, rock salt, perovskite, and amorphous structures), the determination of active sites remains experimentally challenging[12,15,17,19]. The issue is further complicated by the dynamical and heterogeneous surface reconstruction and concomitant

properties evolving[12,22–28]. Thanks to the development of advanced in situ analysis techniques, a consensus has been largely reached that metal oxyhydroxides are the real active phase for OER. Understanding the OER process on them is, therefore, from both fundamental and practical points of view, imperative[29–31]. Unfortunately, the majority of current investigations have been limited to amorphous phases[15,17,21], precatalysts (Co$_3$O$_4$[23], Co(OH)$_2$[12,32]), and analogs (LiCoO$_2$[19]). The evolving structures and properties may obscure insight into the reactivity of the real active sites and the bonding geometries of these materials.

Lateral surfaces or edges of oxyhydroxides are mostly regarded as the active sites, as they correspond to high-index facets that usually hold favorable atomic arrangement, termination, and coordination for efficient electrocatalysis[13,29,33,34]. This consensus has been challenged by the recently proposed mechanisms involving bulk electrocatalysis or an anionic redox process[14,30,35–38], which suggest that basal planes of oxyhydroxides and layered double hydroxides (LDHs) could be responsible for the experimentally observed OER activity. A recent theoretical study showed that the basal plane activity might be on par

[1]State Key Laboratory of Metal Matrix Composites, School of Materials Science and Engineering, Shanghai Jiao Tong University, Shanghai 200240, China. [2]China-UK Low Carbon College, Shanghai Jiao Tong University, Shanghai 201306, China. [3]Department of Chemistry, National Taiwan University, Taipei 10617, Taiwan. [4]National Synchrotron Radiation Research Center, Hsinchu 30076, Taiwan. ✉e-mail: songfang@sjtu.edu.cn

with that of lateral planes[39]. Additionally, the reactivity of basal planes was experimentally suggested for an archetypical OER catalyst of NiFe LDHs[40]. Thus far, the real active sites of oxyhydroxides are in debate.

Here, we identify the geometric catalytic active sites of crystalline oxyhydroxides OER catalyst by investigating the correlation between structural parameters and catalytic activities. To bypass the dynamic surface reconstruction, we choose crystalline CoOOH as the model catalyst, because they were previously demonstrated to be catalytically and structurally stable. The size- and morphology-dependent catalytic activities suggest a ~50-fold difference in specific activities between nanorods (NR) and nanosheets (NS). To understand the link between facets and relativities, we deposit CoOOH NR and NS on electrodes at submonolayer mass loadings (0.1–2 µg/cm²), whose surface areas are subsequently quantified with high-throughput scanning electron microscopy, and their activities are recorded for structure-performance correlation analysis. We unveil a linear correlation between lateral facet areas and catalytic activities, which explicitly supports the view that lateral facets rather than basal facets are catalytic active for OER catalysis. Density functional theory (DFT) calculations are performed to uncover the origin of the reactivity of lateral {10$\bar{1}$0} and {1$\bar{1}$00} facets and the root cause for the inertness of basal {0001} facets. In addition, the influence of oxygen vacancies is investigated. Our work aims to provide experimental evidence for controversial active site determination and to set a basis for understanding facet-depending activities of hydroxides-based OER catalysts.

## Results

### Synthesis of CoOOH NR and NS

Topotactic conversion is the most prevailing approach to produce CoOOH (reaction: $Co(OH)_2 + OH^- \rightarrow CoOOH + H_2O + e^-$). To release the micro-stress of lattice shrinkage from $Co(OH)_2$ to CoOOH, we developed a mild topochemical oxidation approach to synthesize the well-defined CoOOH NR and NS (see details in Methods). The oxidizing condition was carefully optimized to minimize surface distortions and cracks. We found that NaClO was a preferred oxidizing agent when being dissolved in a concentrated KOH solution at 50 °C. The basal sizes of the CoOOH NS were tuned by regulating the growth of precursory $Co(OH)_2$ nanoparticles, in order to enable the correlation analysis (Supplementary Figs. 1 and 2 and Supplementary Note 1). To obtain highly uniform-sized NS, $Co(OH)_2$ nanoparticles were subjected to differential centrifugation before the topochemical oxidation (Supplementary Note 2). For size-dependent and correlation analysis, four lateral sizes ranging over one order of magnitude were synthesized for CoOOH NS, denoting NS-S (small nanosheets: 0.10 ± 0.04 µm), NS-M (medium nanosheets: 0.5 ± 0.08 µm), NS-ML (medium-large nanosheets: 1.2 ± 0.3 µm), and NS-L (large nanosheets: 3.7 ± 1.4 µm). The particle sizes were averaged from measurements over around 100 particles (Supplementary Table 1).

### Microstructures of CoOOH NR and NS

Powder X-ray diffraction patterns (Supplementary Fig. 3) and Raman analysis (Supplementary Fig. 4) suggest the overall topochemical conversion of $Co(OH)_2$ to CoOOH (hexagonal rhomb-centered crystal structure, space group of $R\bar{3}m$, JPCD: 01-073-0497). CoOOH NS have a well-defined hexagonal shape and a uniform size, as revealed by scanning and transmission electron microscopies (SEM and TEM, Fig. 1a, b and Supplementary Fig. 5) and atomic force microscopy (AFM) (Supplementary Fig. 6). The hexagonal SAED pattern (inset in Fig. 1b) suggests the single crystal nature of each hexagonal particle. The lattice spacing of 0.24 nm was observed in the high-resolution TEM (HRTEM) image (Fig. 1c) and corresponds to {10$\bar{1}$0} planes of crystalline CoOOH. These data suggest that the hexagonal crystals are enclosed by basal {0001} facets and lateral {10$\bar{1}$0} facets. Although the four CoOOH NS exhibit a substantial difference in basal sizes (ranging

from ~100 nm to ~5 µm), they have a similar thickness of ~60 nm (Fig. 1j, k, Supplementary Figs. 5 and 6, and Supplementary Table 1). AFM images manifest a low roughness for CoOOH NS ($R_a < 1.5$ nm), which indicates few distortions or cracks are present on basal planes (Supplementary Fig. 6 and Table 2).

The microstructures of CoOOH NR are shown in Fig. 1d−i. They can be regarded as NS stacked along with <0001> direction with a thickness of 50–170 times higher. The basal size is around 800 nm, between those of NS-ML and NS-M. The hexagonal SAED pattern (right-top inset in Fig. 1e) and the clear lattice fringe corresponding to (0001) planes (Fig. 1f) suggest that NR preferentially grows along <0001> direction, exhibiting lengths of around 3–10 µm (Fig. 1d, k and Supplementary Table 1). The PXRD pattern of CoOOH NR shows much higher (012) and (015) reflections over that of NS (Supplementary Fig. 3c, d), in line with the preferential crystal growth. To illustrate the detailed lateral microstructure, CoOOH NR was embedded in epoxy resin and cut to ultrathin slices for TEM observations (Fig. 1g). CoOOH NR has similar hexagon-shaped (0001) basal facets as that of CoOOH NS, and it is also laterally enclosed by {10$\bar{1}$0} facets (Fig. 1h, i). In addition, CoOOH NR shows a Brunauer-Emmett-Teller (BET) surface area of 2.6 m² g⁻¹, which is only ~5% of that of CoOOH NS. The substantial difference is attributed to the long-range stacking of basal planes along the <0001> direction of CoOOH NR. Using the standard crystal structure model of CoOOH, we can calculate a value of 7%, which is close to the experimental data of 5% (Supplementary Fig. 7).

To obtain further structure information, X-ray absorption spectroscopy (XAS) was conducted on CoOOH NR and NS together with reference samples of $Co(OH)_2$, Co foil, CoO, $Co_3O_4$, and $Co_2O_3$. Figure 2a presents the normalized cobalt K-edge X-ray absorption near-edge spectra (XANES). The absorption edge of the main resonance of the Co K-edge is used to determine the Co valence. Based on the first derivative of the main absorption edge, a linear relationship of the oxidation states against the main absorption edge is exhibited (Fig. 2b). Both CoOOH NR and NS have a near +3 valence state of cobalt. The electron energy loss spectra (EELS, Fig. 2d) of Co L-edge show two characteristic white lines of $L_2$ and $L_3$. They are attributed to the electron transition from the spin-orbit split $2p_{3/2}$ and $2p_{1/2}$ levels to the available states in the $3d$ band. The intensity ratio of $L_3/L_2$ is directly related to the valence state of Co. Both NR and NS have $L_3/L_2$ ratios of around 3, close to the reported value for $Co^{3+}$ [41]. The high-resolution X-ray photoemission spectra (XPS) of Co $2p$ show characteristic peaks of $Co^{3+}$ (Fig. 2g)[32,42,43].

Extended X-ray absorption fine structure (EXAFS) $k^2\chi(R)$ spectra of cobalt K-edge (Fig. 2c) were employed to probe the local geometry of cobalt. There are two scattering paths around the absorbing Co ions of CoOOH NR and NS. The first peak located at ~1.44 Å is attributed to the first coordination shell of the Co-O scattering path and the second one located at ~2.45 Å could be ascribed to neighboring cobalt metals (that is, Co−Co) surrounding the absorbing Co ions. The shorter Co-O and Co-Co paths over those of $Co(OH)_2$ agree well with the full oxidation of $Co(OH)_2$. The absence of the Co-Co path located at 3.1 Å rules out the formation of $Co_3O_4$. Wavelet transform (WT) EXAFS analysis (Fig. 2e, f) reveals that the Co-O paths slightly shift to a higher k space in NR. This probably results from the dominating dangling Co-O bonds on the surface of NR, in sharp contrast to $\mu_3$-O sites as the primary species on the surface of NS (inset in Fig. 2e, f). The different surface Co−O coordination detected above corresponds well with the distinct exposed facets of CoOOH NR and CoOOH NS[44,45].

To further understand the structural parameters, the average coordination numbers for CoOOH NR and NS were quantitatively analyzed involving a simulation of $k^3$-weighted EXAFS spectra (Supplementary Fig. 8 and Supplementary Table 2). CoOOH NR exhibits the coordination number (CN) of ~5.7 for the Co-O path, corresponding with the typical octahedral-coordinated Co-O bonding of CoOOH.

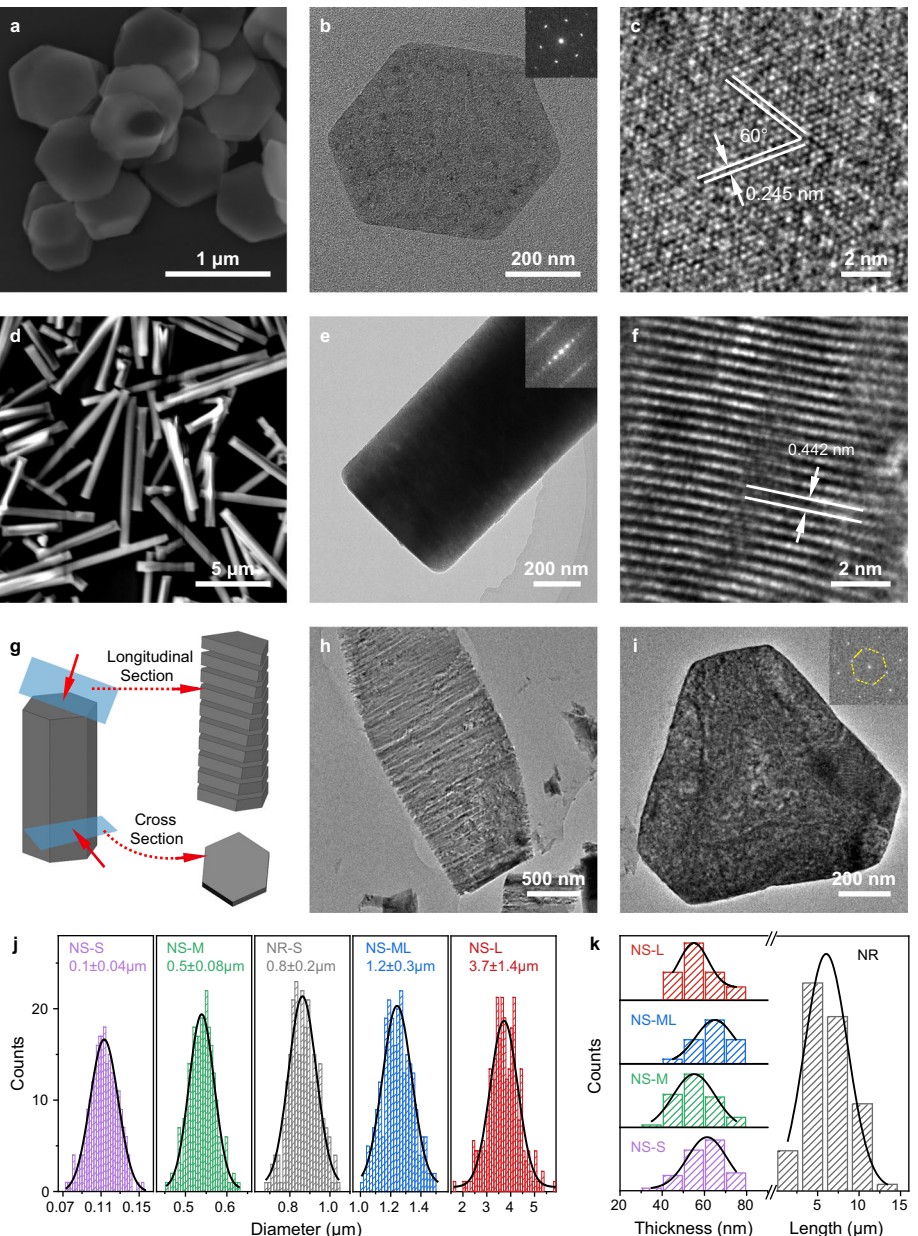

**Fig. 1 | Microstructures of CoOOH NS and NR. a, d** SEM images, **b**, e TEM images, and **c**, **f** HRTEM images of CoOOH NS-M (**a–c**) and NR (**d–f**); **g** Schematic illustration of the sectioning of CoOOH NR for TEM observation; **h**, **i** TEM images of CoOOH NR after sectioning perpendicular (**h**) and parallel (**i**) to the basal surface. The NR in (**h**) disintegrated, and it should result from the shear stress in the sectioning procedure; **j** Basal plane size distribution of NS and NR. Standard errors were calculated from the standard deviations of the sizes of all the observed nanoparticles; **k** Thickness/height distribution of NS and NR. The particle sizes were based on measurements over an average of around 100 particles. The thicknesses of NS were measured by AFM, while other dimensions were measured by high-throughput SEM. Source data are provided as a Source Data file.

CoOOH NS shows a lower peak intensity, corresponding to a CN of 5.3 for the Co-O path. The lower CN of CoOOH NS should be attributed to the larger amount of oxygen vacancies on basal (0001) facets. The formation of oxygen vacancies might release electrons back to Co $3d$ orbital, thereby lowering the valence of metal ions to some extent. Careful analysis of the Co L-edge EELS shows that NS has a slightly higher $L_3/L_2$ ratio than NR, in line with the lower Co valence on the surface of NS. Additionally, the energy loss near-edge fine structure of oxygen K-edge exhibits a weakened pre-peak at around 536 eV for CoOOH NS. The pre-peak is attributed to the electron transition from O $1s$ to O $2p$ fractions in unoccupied states hybridized with the Co $3d$ orbitals. The lower pre-peak suggests the higher occupancy of Co $3d$ orbitals, corresponding well with the lower Co valence and the

presence of more oxygen vacancies in NS[46]. XPS analysis on O $1s$ shows that CoOOH NR has a stoichiometric $O_{lat}$/OH atoms ratio of 0.95, whereas the ratio of CoOOH NS is only 0.81 (Fig. 2h). This should be ascribed to the water dissociation on basal oxygen vacancies of CoOOH NS[13].

Taken together, both CoOOH NR and NS exhibit octahedrally coordinated $Co^{+3}$ ions, which pack in an edge-sharing manner to form crystalline β-CoOOH. Facets of $\{10\bar{1}0\}$ are the dominating surface for CoOOH NR, whereas facets of $\{0001\}$ take the majority of the surface of CoOOH NS. A small portion of oxygen vacancies are present on both facets, and the basal $\{0001\}$ facets contain more. In addition, the full XPS spectra rule out the presence of Fe impurities in NS and NR (Supplementary Fig. 9).

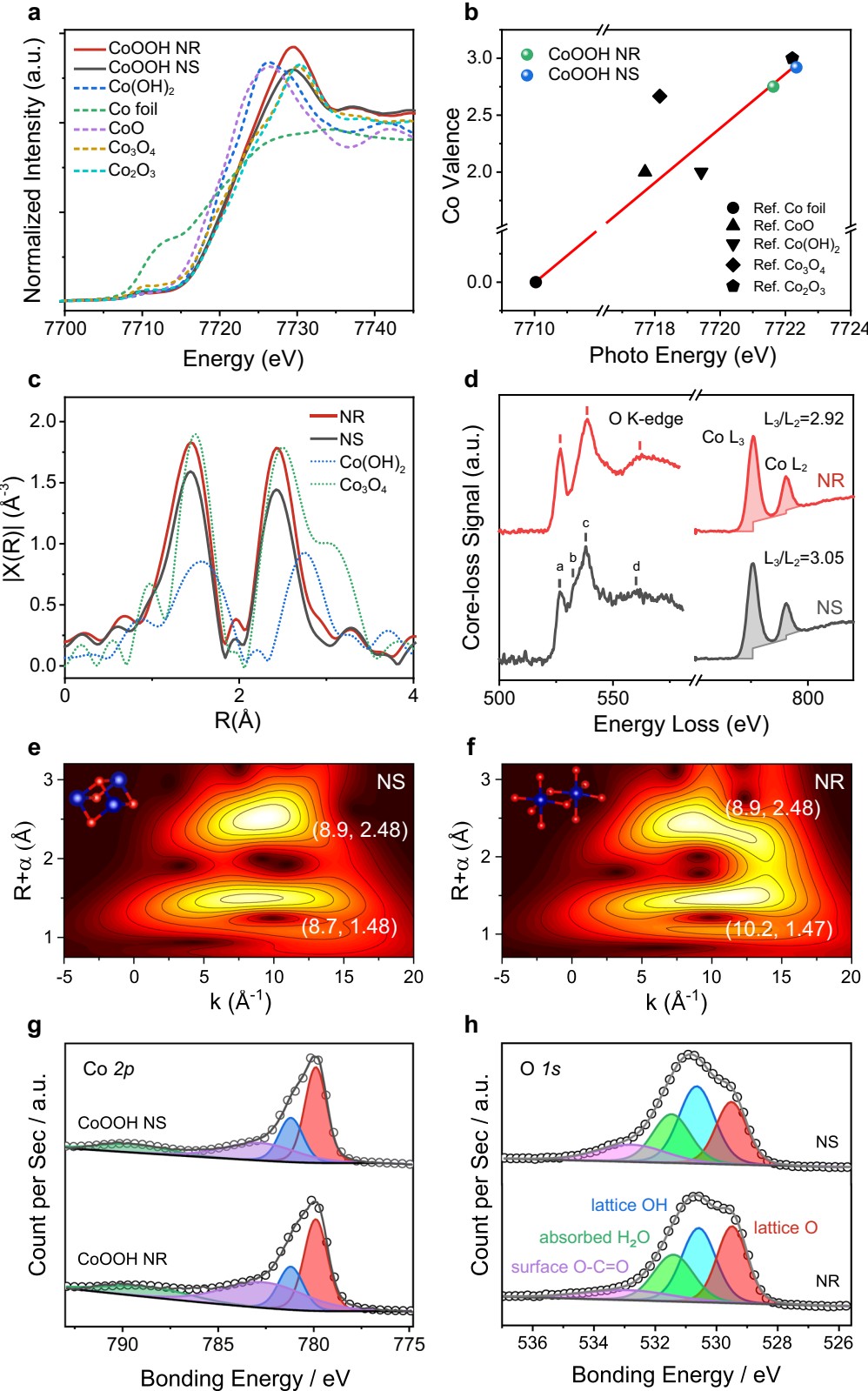

**Fig. 2 | X-ray absorption, electron energy loss, and X-ray photoemission spectra of CoOOH NS and NR. a** XANES spectra of cobalt K-edge for NR (red) and NS (black). Reference samples of Co(OH)$_2$ (blue), Co foil (green), CoO (purple), Co$_3$O$_4$ (yellowish-brown), and Co$_2$O$_3$ (purple) are included; **b** Fitted oxidation states of Co for NR (green point) and NS (blue point); **c** Fourier transform (FT) of Co

K-edge EXAFS for NR (red) and NS (black); **d** EELS spectrum of oxygen K-edge and Co L-edge of CoOOH NR (red) and NS (black); **e, f** WT-EXAFS plots of **e** NS and **f** NR; **g, h** XPS spectra of **g** Co 2*p* and **h** O 1*s*. Source data are provided as a Source Data file.

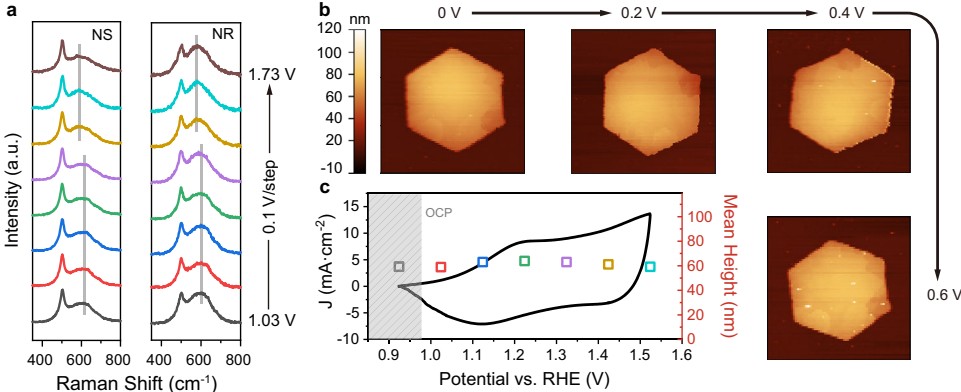

**Fig. 3 | In situ Raman and EC-AFM analysis of CoOOH NS and NR. a** Raman spectra under applied potentials ranging from 1.03 to 1.73 V vs. RHE (from bottom to top, with the interval of 0.1 V); **b** AFM images and **c** corresponding height of NS under applied potentials ranging from 1.03 to 1.73 V vs. RHE. The corresponding CV curve was also shown in (**c**). AFM measurement was interfered with by the evolving bubbles at potentials higher than 1.53 V vs. RHE. Source data are provided as a Source Data file.

## Local chemical and structural stability of CoOOH NR and NS in OER

In situ Raman and electrochemical atomic force microscopy (EC-AFM) analyses were conducted to examine the structural stability under potential for OER, prior to the detailed electrochemical analysis. Both CoOOH NR and NS show characteristic Raman peaks of β-CoOOH, that is, an $A_{1g}$ vibrational mode of Co–O located at 602 cm$^{-1}$ and an $E_g$ vibrational mode at 508 cm$^{-1}$ (Fig. 3a). The higher $A_{1g}$ peak of CoOOH NR should be ascribed to the increasing ordering along <0001> direction. The $A_{1g}$ peak shifted from 602 to 578 cm$^{-1}$ under potentials over 1.43 V vs. RHE. This is commonly attributed to the oxidation of a small fraction of Co(III) to Co(IV)[47,48]. The partial oxidation does not change the local structures, as previously evidenced by several groups using an advanced technique of in situ XAS[16,49]. In line with this, EC-AFM images (Fig. 3b, c) show little change in dimensions (both thickness and basal size). This is contradicted by the recent report where Co(OH)$_2$ exhibited ion-(de)intercalation-induced height change during the catalytic OER[12]. The discrepancy should be due to the distinct starting catalysts (Co(OH)$_2$ in their research and CoOOH here for us). Source data are provided as a Source Data file.

## Correlating catalytic activities with structure parameters

The structure-activity correlation was first investigated at mass loadings of dozens of micrograms. The catalytic activities of the CoOOH NS and NR were evaluated towards OER by steady-state electrochemistry measurements in 1 M KOH solutions using a typical three-electrode cell setup. To alleviate the conductivity issue of transition metal oxide electrocatalysts, acetylene black was mixed with catalysts before electrochemical characterization.

We investigated the size-dependent activities of CoOOH NS in the beginning. The polarization curves in Fig. 4a show that the smaller the size, the higher the activity. To deliver a current density of 10 mA cm$^{-2}$, CoOOH NS-S, NS-M, NS-ML, and NS-L require overpotentials of 426, 441, 462, and 492 mV, respectively. All CoOOH NS share, regardless of their varying sizes, a Tafel slope of ~60 mV (inset in Fig. 4a). The Tafel slope is a kinetic parameter, which has an implication on the catalytic mechanism. Though the interpretation is controversial (in particular for reaction coupled with multiple electron transfer), a Tafel slope of 60 mV could be associated with a catalytic mechanism involving chemical reaction as the rate-determining step after the first electron transfer[16]. Given the same kind of precursory materials (Co(OH)$_2$) and the same topochemical oxidation process for materials synthesis, it is rather fair to assume the samples possess similar surface electronic structures and states and thus proceed with the OER via the same catalytic mechanism. In addition, the current densities were recorded at a relatively lower overpotential, which suggests the coverage of the adsorbed species should be low, thereby allowing the Tafel analysis in conjunction with the Butler–Volmer equation[50]. The size-independent mechanism and the low coverage of the adsorbed species guarantee a quantitative correlation analysis of catalytic activities with geometric surface areas. Gas chromatography experiments, drainage method, and rotating ring-disc electrode (RRDE) method revealed Faradaic efficiencies of >95% for CoOOH NS and NR, suggesting that the anodic currents were exclusively for the OER catalysis (Supplementary Fig. 10). Current densities at an overpotential of 425 mV ($J_{@\eta=425\,mV}$) are chosen to measure the activities. The reason for this choice is that 425 mV are located in their respective Tafel region (Fig. 4a), where the current densities are not interfered with by mass transport and thus justify the comparison. The current densities of NR and NS are dependent on the intrinsic activity and the surface area of each specific plane (basal and lateral plane here). They are calculated from Eq. 1,

$$J_{overall} = J_b + J_l = j_b \times S_b + j_l \times S_l = j_b m/\rho h + 8 j_l m/\sqrt{3}\rho d \qquad (1)$$

where $J_{overall}$, $J_b$, and $J_l$ are the overall, basal, and lateral current density, respectively; $j_b$ and $j_l$ are the specific current densities on basal and lateral planes, respectively; $S_b$ and $S_l$ are the basal and lateral surface area, respectively; m and ρ are the mass and density of catalysts, respectively; h and d are the height and basal size of NS, respectively (detailed discussion is shown in Supplementary Notes 3, 4, 5). Given the same values of $j_b$, $j_l$, m, ρ, and h for all the samples, the equation indicates that their overall current densities are proportional to the reciprocal basal size (1/d). Figure 4b shows the experimental correlation of current densities with the reciprocal basal sizes. It follows a power-law exponent relationship ($J = J_0 + k \times d^{-n}$, see discussion in Supplementary Note 5) and gives a fitting power of around 0.5. Ideally, the correlation could indicate the geometric location of catalytic active sites (Fig. 4b and Supplementary Note 5): for $n = 1$ (activities are proportional to the reciprocal basal sizes), active sites are located only on lateral facets if $J_0 = 0$ (corresponding to $j_b = 0$ and $j_l \neq 0$ in Eq. 1) and on both lateral and basal facets if $J_0 \neq 0$ (corresponding to $j_b \neq 0$ and $j_l \neq 0$ in Eq. 1); for $n = 0$ (activities are independent on the sizes of basal facets), active sites are on basal facets only (corresponding to $j_b \neq 0$ and $j_l = 0$ in Eq. 1). The fitting power of $n = 0.5$ indicates the reactivity of lateral facets but cannot tell whether the basal facets are active or not. The deviation of fitting power could be attributed to the particle agglomeration, which is inevitable at mass loadings of dozens of micrograms and results in electrically unavailable regions.

To further test the reactivity of the basal surface, the morphology-dependent activities were studied on NR and NS,

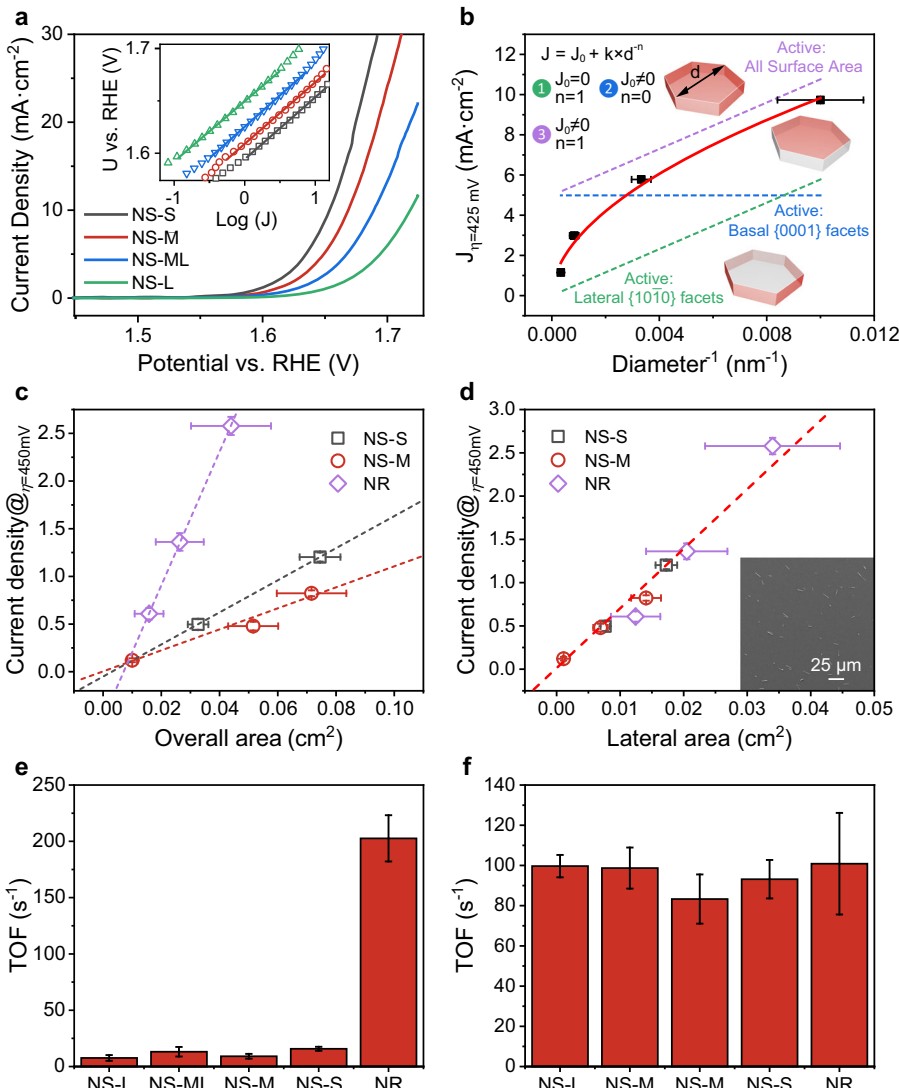

**Fig. 4 | Catalytic activities of CoOOH NR and CoOOH NS for oxygen evolution reaction. a** Polarization curves of NS-S (black), NS-M (red), NS-ML (blue), and NS-L (green). The inset shows the corresponding Tafel plots; **b** Size-dependent activities of NS at the same loadings; **c, d** Correlation of the current densities with **c** the overall surface areas and **d** the lateral facet areas. NS-S is displayed in black, NS-M in red, and NR in purple. The inset in (**d**) is the SEM image of NR spin-coated carbon electrodes for statistical analysis. **e, f** TOFs at the overpotential of 450 mV when assuming cobalt atoms on (**e**) basal facets or (**f**) lateral facets are active. Standard errors for activities were calculated from the standard deviations from three measurements, and that for areas were calculated from the standard deviations of the measured particle sizes. Source data are provided as a Source Data file.

considering their substantial difference in basal and lateral surface areas. To simplify the correlation, NS-ML was chosen because its basal size is similar to that of NR (NR: 0.8 μm; NS-ML: ~1.2 μm). Electrochemical measurements reveal that CoOOH NR is two to three times more active than NS-ML (Supplementary Fig. 11). The specific activities could be calculated by normalizing the catalytic current density with BET surface area. Owing to the substantial difference (~20 times) in geometrical surface areas, CoOOH NR exhibits a ~50 times higher specific activity than NS. As stated before, the identical Tafel slope (around 60 mV dec$^{-1}$, Supplementary Fig. 12) excludes the contribution from the different mechanisms and mass/charge transportation. The observed substantial difference in specific activities could be well rationalized with the facet-dependent activities. As CoOOH NR and NS have a similar basal size, the lateral facets show only a 1.5 times difference in areas despite their substantial difference in overall surface area. The minor difference (1.5 times) corresponds well with the two to three times difference in geometrical activities. The coincidence suggests that the lateral facets

rather than basal facets are active for OER. The reactivity of lateral facets is further confirmed by their similar TOF$_{EIS}$ extracted from the EIS, which is a measure of intrinsic activity (Supplementary Fig. 13 and Supplementary Table 3)[51].

To confirm the facet-dependent activity, we fixed the loading of CoOOH NR at 0.285 mg cm$^{-2}$ and varied loadings of NS to reach the same overall surface area, basal facets area, and lateral facets area as that of NR, respectively. The loading was calculated based on the surface area estimation by statistically counting the particles' size in TEM, SEM, and AFM observations (Supplementary Note 4). CoOOH NR shows significantly higher activity than NS when they have the same overall or basal facet area. Notably, their activities are comparable in the same lateral facet area (Supplementary Fig. 11d). This strongly supports the reactivity of the lateral facets and the inertness of the basal facets. The lateral facet reactivity was further validated by the size-dependent activities of CoOOH NR, where a perfect linear correlation of catalytic activities with the reciprocal of the basal radius is present (Supplementary Fig. 14).

Correlating activities with structural parameters, in particular, for those with distinct well-defined shapes and at submonolayer loadings, can give explicit evidence for the identification of real geometric active sites of catalysts[13,21,52,53]. This approach was employed here to verify the reactivity of the lateral facets and the inertness of the basal facets. A mixed solvent dispersion protocol followed by a spin-coating deposition process was developed, to well disperse CoOOH on electrodes at submonolayer loading (see details in Methods). The coverages were varied by regulating the catalysts' concentrations. In general, two to three coverages were employed for each sample. High-throughput SEM was conducted to evaluate the discreteness and count the real surface area for statistical analysis (inset in Fig. 4d, see details in Supplementary Note 4). Typical SEM images show that CoOOH NR and NS are discretely dispersed, showing few aggregations or overlaps.

The polarization curves were recorded at two to three loadings for each kind of sample (Supplementary Fig. 15). The current densities at an overpotential of 450 mV were used to represent the activities. They were first correlated with their overall surface areas. For each sample, the activities increase linearly with surface areas (Fig. 4c, d and Supplementary Fig. 16). The linear relation suggests that the electrochemical analysis eliminated the interference from mass and charge transport at submonolayer loadings owing to the discrete dispersion of nanoparticles. It is worth noting that CoOOH NR exhibits a linear slope of 3.7–6.4 times that of NS and the linear slopes of NS are size-dependent (Fig. 4c and Supplementary Fig. 16a, b). The morphology- and size-dependent linear slopes confirm the facet-dependent activity, that is, lateral facets (dominating in CoOOH NR) are much more active than basal facets (dominating in CoOOH in NS). To elucidate the role of each specific facet, we correlated the current densities with lateral and basal surface area, respectively. Impressively, the activities of all the samples, regardless of the sizes and shapes, show a strong correlation with the lateral facet areas, rather than the basal surface areas. (Fig. 4d and Supplementary Fig. 16c). The perfect linear fitting strongly supports the reactivity of lateral facets and the inertness of basal facets. Accordingly, CoOOH NR shows a TOF close to that of NS (around 90 s$^{-1}$) when cobalt atoms of lateral facets are assumed active (Fig. 4e, f and Supplementary Fig. 17). Additionally, we compared our samples with the reported cobalt hydroxides at a similar size, which showed comparable TOFs (Supplementary Table 4).

Furthermore, we linked the activity with the chemical information of CoOOH nanoparticles in the catalysis process, to verify the facet-dependent activities. Previous mechanism analysis revealed that the catalysis of $CoO_x$ was associated with the intermediates of "active oxygen" species[16,54,55]. The in situ Raman analysis on CoOOH NR and NS shows a vibrational mode with a double-peak feature located at ca. 1165 and 1021 cm$^{-1}$ (Supplementary Fig. 18). They are prevailingly assigned to the well-known "active oxygen" species that have the superoxide nature[56,57]. Under the OER condition, the peak intensity remains constant, in line with the intermediate nature of the corresponding species for OER catalysis. To quantify the concentration of "active oxygen" species, the peak ratio of $I_{1165}/I_{502}$ was calculated, where $I_{502}$ is the peak intensity of the reference $E_g$ vibration mode of Co(III)-O. CoOOH NR has a peak ratio of $I_{1165}/I_{502} = 0.24$, which is nearly twice higher than that of CoOOH NS ($I_{1165}/I_{502} = 0.11$), This, from a molecular point of view, supports the reactivity of lateral planes (Supplementary Fig. 18).

### Theoretical calculation of the geometrical active sites of CoOOH

To elucidate the catalytic active sites, we first scrutinized the difference in the coordination of surface oxygen atoms. Basal (0001) facet and lateral (10$\bar{1}$0) and (1$\bar{1}$00) facets were investigated (Fig. 5a, left panel). The reason for choosing the former two facets is that they are the dominating facets observed by TEM. The consideration of the additional lateral (1$\bar{1}$00) facets is because they could be formed at the truncated edges or present as nanoscale steps on lateral (10$\bar{1}$0) facets. As shown in Fig. 5a, basal (0001) facets are terminated by fully

coordinated sites (Co$_{FCS}$) of cobalt atoms, bonded by threefold coordinated oxygens (O$_{3c}$). In contrast, lateral (10$\bar{1}$0) facets have coordinatively unsaturated sites (Co$_{CUS}$), linking with four neighboring equatorial twofold-coordinated oxygen atoms (O$_{2c}$) and one apical O$_{3c}$ site. Lateral (1$\bar{1}$00) facets contain both Co$_{FCS}$ and Co$_{CUS}$, which can be regarded as a hybrid of the former two facets.

We then investigated the electronic structures by performing density functional theory calculations at the PBE+U+D3 level (Fig. 5a–d, Supplementary Figs. 19 and 20, and Supplementary Table 5). The Co$_{CUS}$ were considered active sites for lateral facets, as adsorbates are prone to bind on to form Co-O$_{1c}$ along the (10$\bar{1}$0) apical or (1$\bar{1}$00) equatorial direction (Fig. 5a, left panel). The Co atoms on lateral facets exhibit higher d-band centers compared with that on the basal (0001) facet (Supplementary Fig. 20), implying their favorable absorption of reaction intermediates[58,59]. The oxygen 2$p$-band center was calculated (O$_{1c}$ of lateral facets and O$_{3c}$ of basal facets) because it reflects the metal d-character more accurately and has a direct relation to the catalytic reactivity[60]. The lateral facets upshift the O 2$p$-band center closer to the Fermi level over that of the basal facets, accompanied by a reduced energy gap between the Co 3$d$ and O 2$p$-band centers (Fig. 5b and Supplementary Fig. 20). The change in electronic structures indicates a greater covalency of Co-O bonds on lateral surface[14,60]. In line with this, the surface-adsorbed Co-O bond shows much less electron localization function (ELF) values (0.52 and 0.54) than that of lateral facets (0.60), suggesting a less polarized and less ion-featured Co-O bond (Fig. 5a, right panel)[61]. The Co-O bond strength (Co$_{CUS}$-O$_{1c}$ for lateral facets and Co$_{CUS}$-O$_{3c}$ for basal facet) was investigated using crystal orbital Hamilton population (COHP) analysis (Fig. 5c). The integrals of COHP up to the Fermi level for lateral facets (3.70 and 3.71) are higher than that for basal facets (1.87), verifying the stronger Co-O bond on lateral facets. In addition, lateral facets have a short Co-O bond length and a higher bond dissociation energy (Supplementary Table 5). The stronger Co-O bond rationalizes the less oxygen vacancies probed on CoOOH NR than that on NS. According to previous reports[14,62], the hybridization configuration associated with a stronger metal-oxygen bond (Fig. 5d) manifests a higher reactivity of oxygen atoms and a favorable catalytic kinetics for oxygen evolution reaction.

The DFT calculations were then performed to gain in-depth insight into the catalytic mechanism of CoOOH nanocrystals at the PBE+U+D3 level (Fig. 5e–j, Supplementary Figs. 21–28, and Supplementary Table 6). Given the experimental inertness of basal planes, the recently proposed mechanisms involving bulk reactivity or anionic redox[14,30,35–38] may not proceed in our crystalline CoOOH. In this regard, the OER reaction ($2H_2O \rightarrow O_2 + 4(H^+ + e^-)$) is assumed to process through a conventional proton-coupled electron transfer four-step mechanism, involving *OH, *O, *OOH, and *OO (the asterisk denotes the adsorption site) as reaction intermediates[29,39,63–65] (See details in Supplementary Methods).

Basal (0001) facets show the highest thermodynamic overpotential of 1.95 V. (Fig. 5f–h). The potential-determining step (PDS) is the transition from *O to *OOH (step 3). Several other groups did a similar calculation on basal facets before (For instance, ref. 64 and ref. 29 reported overpotentials of 1.71 and 0.8 V for CoOOH, respectively; Cui showed an overpotential of 1.26 V for LiCoO$_2$[19]). Despite a certain difference in overpotentials and PDSs, they all suggested the inertness of (0001) planes. Owing to the distinct atom arrangement, the free-energy landscape is altered substantially on lateral (1$\bar{1}$00) and (1$\bar{1}$00) facets (Fig. 5f–j). The PDS shift to the absorption of *O through the deprotonation of *OH (step 2). The thermodynamic overpotentials are lowered by -1.3 V, reaching 0.66 V for lateral (10$\bar{1}$0) facets and 0.70 V for lateral (1$\bar{1}$00) facets (Fig. 5f, i, j, and Supplementary Fig. 28b). The shifting PDS and the decreasing overpotential correspond to a larger $\Delta G_{*O}-\Delta G_{*OH}$, in accordance with the upshifting of O 2$p$ states for lateral facets[66]. Additionally, DFT calculations were also performed without

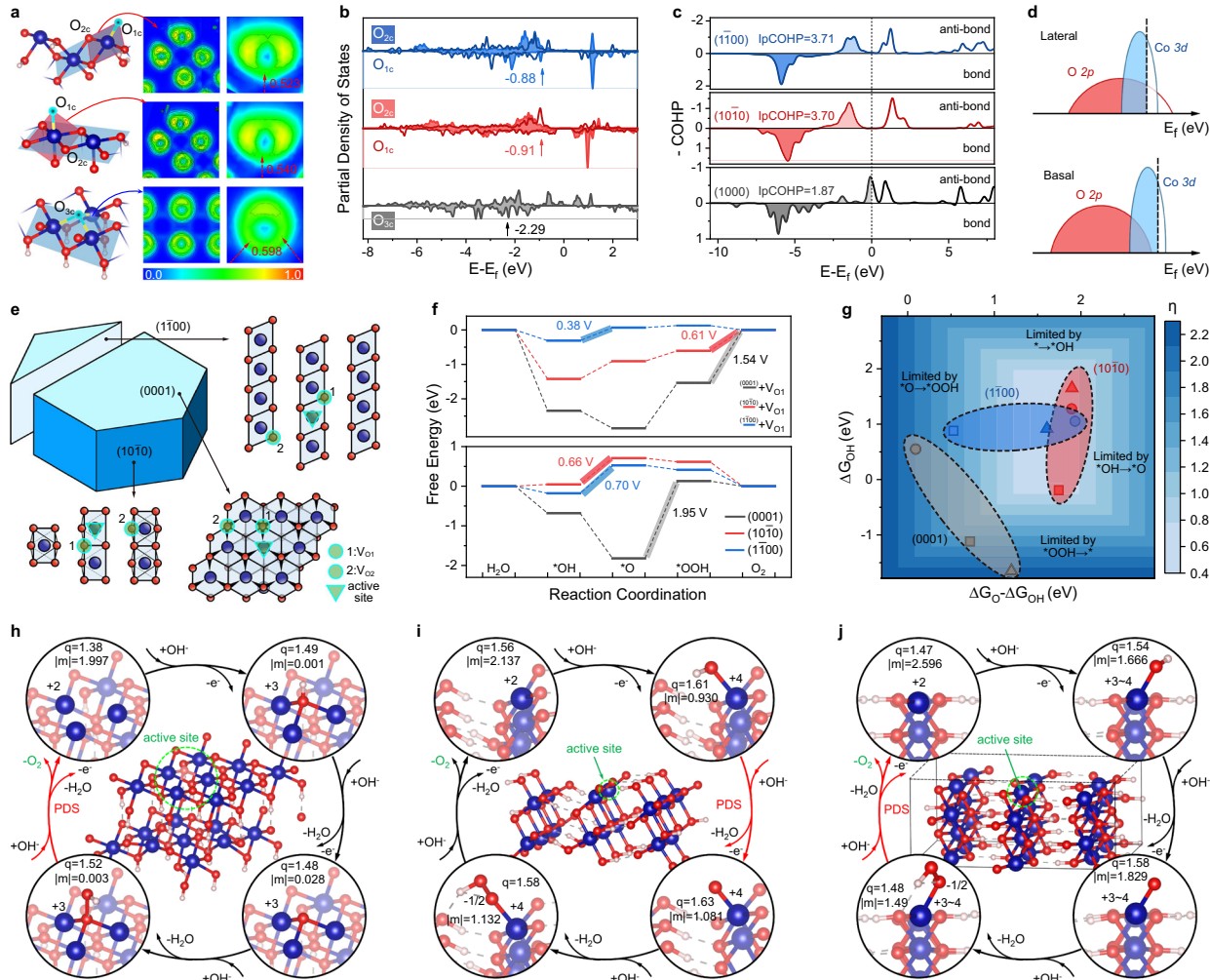

**Fig. 5 | Theoretical calculation of the active sites of crystalline CoOOH.**
**a** Schematics depicting the arrangement of surface atoms on (0001), (10$\bar{1}$0), and (1$\bar{1}$00) facets and their corresponding electron localization function (ELF) contour slices of the surface Co-O bond; **b** The electron density of states for the oxygen $2p$ orbitals of $O_{2c}$/$O_{3c}$ atoms (shaded) and $O_{1c}$ atoms. (1$\bar{1}$00) is displayed in blue. (10$\bar{1}$0) is displayed in red and (0001) in black. **c** Crystal orbital Hamilton populations (COHP) of the surface Co-O bond (Co$_{CUS}$-O$_{1c}$ for lateral facets and Co$_{CUS}$-O$_{3c}$ for basal facets). (1$\bar{1}$00) is displayed in blue. (10$\bar{1}$0) is displayed in red and (0001) in

black. **d** Schematic band diagrams of the basal and lateral facets. **e** Schematics of investigated facets and oxygen vacancies on them. **f** The free-energy landscape of the OER on different facets of CoOOH $R\bar{3}m$ crystal (at the PBE+U+D3 level of theory). **g** 2D map of theoretical overpotentials $\eta$. Cubes are for perfect facets. Triangles and circles are for facets with $V_{O1}$ and $V_{O2}$, respectively. **h**–**j** The proposed OER proceeding on **h** (0001), **i** (10$\bar{1}$0), and **j** (1$\bar{1}$00) facets. Source data are provided as a Source Data file.

the inclusion of +U functional or implicit solvation (Supplementary Fig. 27 and Supplementary Table 5). They show the same activity trend despite some differences in the absolute values. To further understand the facet's dependent activities, we estimated the oxidation states of oxygen atoms and cobalt sites by analyzing the Bader charges and local magnetic moments (Fig. 5h–j and Supplementary Table 7)[29]. The adsorption of *OH involves the oxidation of $Co^{2+}$, resulting in $Co^{+3}$ on basal facets and higher valent $Co^{+4}$ or $Co^{+3-4}$ on lateral facets. The *OOH intermediate exhibits a peroxo configuration (−1) on basal facets, whereas a superoxo configuration (−1/2) on lateral facets. The transition to higher valence leads to a weakened absorption, thereby shifting the PDS from desorption of *O for basal facets to absorption of *O for lateral facets.

Given the presence of oxygen vacancies in both CoOOH NR and NS, we discussed their influence on the reactivity of basal and lateral facets (Fig. 5e–g and Supplementary Figs. 21–26). The oxygen vacancies that are nearest and second nearest to active Co sites ($V_{O1}$ and $V_{O2}$ in Fig. 5e) were investigated. The reaction paths are altered significantly in the presence of oxygen vacancies. The thermodynamic overpotentials and PDS are dependent on the location of

oxygen vacancies (Fig. 5f, g). Although the $V_{O1}$ results in a substantial decrease (0.41 V) of the thermodynamic overpotential of basal facets, it (1.54 V) is still much larger than that of lateral facets. It suggests the inertness of basal facets even though oxygen vacancies are accounted for. Several groups investigated the theoretical possibility of activating the basal facets of NiOOH (similar to CoOOH) using different models[39,67]. Oxygen vacancies, of either $V_{O1}$ or $V_{O2}$, have a minor influence on the reactivity of lateral (10$\bar{1}$0) facets. A minimum overpotential of 0.61 V is observed, which is only 0.05 V lower than that for perfect (10$\bar{1}$0) facets. Additionally, we found that oxygen vacancies can substantially activate the lateral (1$\bar{1}$00) facets (although the facets were not observed in our CoOOH NS and NR), yielding a thermodynamic overpotential of only 0.38 V. The introduction of oxygen vacancies results in an absorption configuration of *OOH with a Co-O-O three-membered ring and greatly lowers the formation energy of *OH by introducing a hydrogen bond with a neighboring hydroxyl group. This finding has an implication for the structural engineering of OER catalysts; for instance, it indicates the potential of combining defect engineering and facet regulation to optimize the electronic structure for fast OER kinetics.

Lastly, the calculated adsorption energies for both perfect and imperfect facets were plotted as a scaling relation between $\Delta G_{OOH}$ and $\Delta G_{OH}$ (Supplementary Fig. 28a). It gives a relationship of $\Delta G_{OOH} = 0.69\Delta G_{OH} + 3.20$ eV ($R^2 = 0.95$), corresponding well with reported scaling plots of transition metals oxides[68,69]. Basal and lateral facets (no oxygen vacancy) are located on the left and right legs of the established volcano plot, respectively (Supplementary Fig. S28b). They are more than 0.28 V away from the volcano's top site. The introduction of oxygen vacancies further regulates the reactivity under the traditional scheme for OER catalysis, capable of pushing the active sites closer to the volcano top site as suggested by the imperfect lateral $(1\bar{1}00)$ facets in our calculation (Supplementary Fig. S28b).

## Discussion

In summary, we identified the catalytic active sites of crystalline CoOOH experimentally. We showed that the catalytic active sites were exclusively located on lateral $\{10\bar{1}0\}$ surfaces of CoOOH, as demonstrated by the size- and facet-dependent activities and, in particular, the linear correlation of activities with lateral $(10\bar{1}0)$ surface areas. Theoretical calculation elucidates the inertness of basal $\{0001\}$ facets (1.95 V) and the reactivity of lateral facets of $\{10\bar{1}0\}$ (0.66 V). The reactivity of lateral facets is ascribed to the presence of coordinatively unsaturated cobalt sites, which upshift the O $2p$-band centers closer to the Fermi level and thus enhance the hybridization with the Co $3d$ band. Oxygen vacancies generally lead to further surface activation, which may push the active sites to the top of conventional volcano plots. The disclosing of facet-governing reactivity helps to clarify the fundamental aspects of oxyhydroxide-based OER catalysts and paves the way toward the rational design of next-generation electrocatalysts. To further improve the catalytic activity, we may focus on exposing lateral facets or figure out how to activate the inert surfaces. In addition, combining strategies of defect engineering and facet regulations might be an alternative approach to obtaining exceptional CoOOH catalysts based on the theoretical calculation. The establishment of model catalysts of well-defined CoOOH crystals may set a basis for the mechanism understanding and structural design of efficient metal oxyhydroxide-based catalysts.

## Methods

### General

All chemicals were directly used as they were received from manufacturers. Cobalt(II) chloride hexahydrate (99%), cobaltous(II) nitrate hexahydrate (99%), hexamethylenetetramine (99%), sodium hydroxide (99.99%), potassium hydroxide (99.99%) and $IrO_2$ powder were purchased from Sigma-Aldrich. About 5 wt% Nafion solution was obtained from Macklin. Sodium hypochlorite solution (with 7% available chlorine), ammonia solution (AR, 28–30 wt%), hydrochloric acid, acetone, and alcohol were obtained from Aladdin. Stainless steel mesh was purchased from Goodfellow. Millipore-Q water (18.2 MΩ) was used for all the measurements and synthesis. The mixed standard gas of oxygen and nitrogen for the GC test was provided by Air Liquide.

### Synthesis of Co(OH)₂ nanosheets

Co(OH)₂ nanosheets were synthesized via a co-precipitation method. The size was regulated by varying alkaline agents and their amount in the solution. For the larger-sized nanosheets (L-Co(OH)₂ and ML-Co(OH)₂), hexamethylenetetramine (HMT, 99%, Sigma-Aldrich) was used as an alkaline agent. Typically, 1.5 mmol of cobalt chloride (CoCl₂·6H₂O, 99%, Sigma-Aldrich) was dissolved in 150 mL of deionized water. The solution was heated to 95 °C. After holding the temperature for 10 min, an amount of HMT was added (see Supplementary Table 8), and the mixed solution was refluxed at 95 °C for 3 h under continuous magnetic stirring in an N₂ atmosphere. Pink powder of brucite Co(OH)₂ was collected and washed with deionized water by three-time centrifuging at a speed of 15,000×g for 10 min. For the

smaller-sized nanosheets (M-Co(OH)₂ and S-Co(OH)₂), sodium hydroxide (NaOH, 99%, Sigma-Aldrich) was used. Typically, an amount of NaOH (see Supplementary Table 8) was dissolved in 50 mL of deionized water. 1 mmol of CoCl₂·6H₂O was then added when the solution was heated to 50 °C. The solution was further aged at 70 °C for 1 h under vigorous stirring and N₂ protection. The products were collected and washed with deionized water by three-time centrifuging at a speed of 1800×g for 5 min. The alkaline agents and the amounts used for the corresponding products were listed in Supplementary Table 8.

### Synthesis of Co(OH)₂ nanorods

Co(OH)₂ nanorod was first grown on stainless steel (SS) mesh using a chemical bath deposition technique. Briefly, 5 mmol of cobalt nitrate (Co(NO₃)₂·6H₂O, 99%, Sigma-Aldrich) and 2.5 mmol of ammonium nitrate (NH₄NO₃, >99.0%, Sigma-Aldrich) were added to the mixture of 35 mL of deionized water and 10 mL of 28–30 wt% ammonia (NH₃·H₂O, AR, Aladdin) and stirred in the air for 10 min. The solution was poured into a petri dish and heated at 90 °C for 30 min. After being cleaned in acetone, deionized water, and 3 M HCl for 15 min, and rinsed with deionized water sequentially, a SS mesh was dipped into the precursory solution where the temperature was maintained at 90 °C for 6 h. The Co(OH)₂ nanorods were then dispersed into deionized water by sonification. After being washed with deionized water several times, the products were collected by centrifuging.

### Topochemical conversion of Co(OH)₂ NS/NR to CoOOH counterparts

A chemical oxidation method was adopted for the topochemical conversion of Co(OH)₂ NS or NR to CoOOH counterparts. Briefly, 10 mg of the as-prepared Co(OH)₂ NS or NR was dispersed into deionized water at a concentration of 1 mg/mL. About 300 µL of 0.5 M NaOH solution was added to adjust the pH to the range of 11–13. The suspension was then heated to 50 °C, and 50 µL of sodium hypochlorite solution (NaClO, 6–14% available chloride, Aladdin, AR) was added as an oxidizing agent. The temperature was held at 50 °C for several hours (4 h for NS and 16 h for NR) to ensure the overall conversion.

### Electrode preparation

For size- and facet-dependent activities analysis, electrocatalysts were loaded on the GC electrode (3 mm in diameter). The CoOOH NR has a basal surface area of nearly 100 times that of CoOOH NS-ML and a lateral area of ~1.88 times that of CoOOH NS-ML at the same loading (see Supplementary Note 3). A suspension of ~3.33 wt% K⁺ ion-exchanged Nafion was used as immobilizing binder (see Supplementary Note 6). The catalyst inks of CoOOH were prepared by dispersing CoOOH powder, acetylene black, and K⁺ ion-exchanged Nafion suspension into deionized water and ethanol solution at the ratio of 2:1. The catalyst ink of 5 µL in volume was drop-casted on the GC electrode and dried naturally to form a catalyst thin film. The mass loading of the CoOOH was 0.05 mg cm⁻². The mass ratio of CoOOH:acetylene black carbon:Nafion was 5:1:1. To reach the same basal surface area, lateral surface area, and overall surface area of CoOOH NR at 0.05 mg cm⁻², the mass loadings of the CoOOH NS were modulated to be 0.5 µg cm⁻², 0.094 mg cm⁻², and 0.35 mg cm⁻², respectively. The varying loadings were estimated based on the surface area calculated in Supplementary Note 4.

For the correlation analysis, CoOOH electrocatalysts were spin-coated on flat electrodes, including HOPG and carbon-coated chips. CoOOH was dispersed in formamide (see Supplementary Note 2) and then diluted by DMSO and formamide to obtain stable colloids at a fixed concentration. CoOOH colloids were then spin-coated on HOPG (2 cm × 2 cm) or carbon-coated chip (diameter: 1 cm) at 3200 rpm. The loading of the catalysts can be readily adjusted by repeating

spin-coating several times. Finally, the flat electrodes were rinsed with water and ethanol several times, to remove the residual solvent.

## Electrochemical measurements

Electrochemical measurement was performed in a three-electrode configuration using a VMP3 Multichannel Potentiostat (Bio-Logic, France), where a modified GC electrode or flat electrode was used as working electrodes. All the electrochemical data were obtained under normal pressure and temperature. A platinum foil was used as the counter electrode and Hg/HgO electrode filled with 1 M KOH solution was used as the reference electrode. The electrochemical measurement was conducted in 1 M KOH electrolytes unless stated otherwise. The reference electrode was calibrated against a reversible hydrogen electrode (RHE) prior to the test. All the potentials were measured against this Hg/HgO (1 M KOH) that has a potential of 0.098 V versus the normal hydrogen electrode. The equilibrium potential for oxygen evolution reaction at any given pH is (1.23-0.098-0.059×pH) V versus Hg/HgO (1 M KOH). Ohmic drop correction was performed via the impedance measurement technique (ZIR) program implanted in the potentiostat. The cyclic voltammetry (CV) curves were measured at a scan rate of $1 \, mV \, s^{-1}$. The linear scan voltammograms (LSV) were measured at a scan rate of $0.5 \, mV \, s^{-1}$. Typically, 30 cycles of CVs were performed to activate and stabilize the electrocatalysts. Five scans of LSV were subsequently carried out to record the catalytic activities. Tafel slopes were calculated from the polarization curves at the current densities range of $1–10 \, mA \, cm^{-2}$. Electrochemical impedance analysis was conducted in an angular frequency window between 100 MHz and 100 kHz (corresponding to processes with time constants between 10 s and 10 μs, respectively). The equivalent circuit used for fitting is shown in Supplementary Fig. 16f. The method for TOF calculation is presented in Supplementary Method 1.

The OER Faradaic efficiency was investigated by the rotating ring-disk electrode (RRDE) technique and gas chromatography (GC) survey. The electrocatalysts were loaded on an RRDE (E7R9 Fixed-Disk, Pine Instruments) electrode, which consists of a glassy carbon rotation disk (disk OD = 5.61 mm) and a Pt ring (ring OD = 7.92 mm). The theoretical collection efficiency of RRDE is 37%. The calculation of Faradaic efficiency is as follows:

$$FE(O_2) = \frac{I_r}{I_d N} \qquad (2)$$

where $I_d$ is the disk current, $I_r$ is the ring current, and N is the current collection efficiency of the RRDE.

In the GC test, the working electrode was prepared by drop-casting 0.2 mg NS or NR or $IrO_2$ powder (from Sigma Aldrich) onto carbon fiber paper ($1 \, cm^2$). Chronopotentiometry was applied at a constant current density of -10 mA cm$^{-2}$. $N_2$ was constantly purged at a speed of $5 \, cm^3 \, min^{-1}$ into the anodic compartment, which was connected to the gas-sampling loop of a gas chromatograph.

## Structural and spectroscopic analysis of the catalysts

The morphology and structure of the resulting materials were characterized by SEM (TESCAN MAGNA SEM), TEM (Talos F200X S/TEM FEI), X-ray diffraction (XRD, Brucker D8 DaVinci diffractometer), and XPS (Kratos AXIS Ultra DLD spectrometer). The high-throughput SEM (Navigator-100 FBT) was carried out to characterize the particles dispersed on the plate electrode. The BET surface area was measured by Micromeritics ASAP 2020. Raman spectra were performed on a confocal Raman microscope with a laser wavelength of 532 nm (Renishaw inVia Qontor 27). AFM was performed with a Dimension FastScan Bio (Bruker) by tapping mode. X-ray absorption spectroscopy measurements were carried out at beamline 01C1 at the National Synchrotron Radiation Research Center in Taiwan. EELS experiments were performed using Gatan GIF Tridiem. The Raman data was collected by

Renishaw inVia Qontor Confocal Raman Spectrometer. Detailed descriptions of data acquisition and analysis are provided in Supplementary Methods 2, 3.

## Computational methods

According to the XRD and TEM analysis, β-CoOOH ($R\bar{3}m$ space group) was used as the computational model. The optimized structures, density of states, and the corresponding unit cell parameters of β-CoOOH are shown in Fig. S15. Basal facets (basal plane) of (0001) and lateral facets of (10$\bar{1}$0) and (1$\bar{1}$00) were cleaved for the simulation of the OER process (Figs. S16–19). A vacuum space of 15 Å was added to all the supercells to avoid interactions between periodic layers. All calculations were carried out within the context of periodic density functional theory (DFT), as implemented in the Vienna ab initio simulation package (VASP), version 5.4.4. The core electrons were described by using the projector augmented plane-wave (PAW) pseudopotential. For each facet, we adopted a Gamma-point centered k-point mesh and used ISPIN = 2 to relax the magnetization. It shows a $Co^{3+}$ configuration for all the slab models based on the calculated Bader charges and local atomic magnetic moments (Supplementary Table 7). The cut-off energy of the plane wave was set to 600 eV and the energy convergence criterion for the self-consistent-field (SCF) cycles was set to $10^{-6}$ eV per cell. The effective Hubbard-U parameter (U-J) of 3.47 eV is obtained by recovering the experimental oxidation energy for the reaction $6CoO + O_2 \rightarrow 2Co_3O_4$ (Supplementary Fig. 19c). We also considered the solvation effect by using the implicit solvation method following the VASPSOL protocol. The method for free-energy calculation is described in Supplementary Method 4.

## Data availability

The data generated in this study are provided in the Source Data file. Source data are provided with this paper. The remaining data that support the findings of this study are available from the corresponding author upon request. Source data are provided with this paper.

## Code availability

The computational codes used in the current work are available from the corresponding author upon reasonable request.

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

## Acknowledgements

This work was supported by the National Natural Science Foundation of China (Grant Nos. 51902200), the National High-Level Talent Program for Young Scholars, the Shanghai Science and Technology Committee (Grant Nos. 19ZR1425300 and 22511100400), the Start-up Fund (F.S.) from Shanghai Jiao Tong University, the Program for Distinguished Young Scholars of the State Key Laboratory of Metal Matrix Composites (SKLMMC). We also acknowledge SJTU Instrument Analysis Centre for the measurements. We thank Haofei Wu and Prof. Pan Liu for their assistance with the EELS test.

## Author contributions

F.S. conceived the ideas and led the project. S.W. synthesized the catalysts, did the structural characterizations, and tested the electrochemical activities, with the assistance of Q.J. S.W. conducted the DFT calculations with the contribution from S.J. C.-S.H. and H.-M.C. did the XAS measurements. D.Z. provides resources for material characterizations and theoretical calculations. F.S. and S.W. contributed to the data analysis and wrote the paper, with input from all the other authors.

## Competing interests

The authors declare no competing interests.
