## [Peer Review File · Nature Communications]

Identifying the Geometric Catalytic Active Sites of Crystalline Cobalt Oxyhydroxides for Oxygen Evolution ReactionReviewers' Comments:

Reviewer #1:

Remarks to the Author:

This manuscript reports an interesting experimental study of the OER on CoOOH nanosheets (NSs) and nanorods (NRs). The results show a higher activity on the NRs, suggesting a higher reactivity of the CoOOH (10-10) facet in comparison to the (0001) basal plane. To support their experimental results, the authors performed DFT calculations of the OER energetics, which show a higher overpotential on the (0001) surface than on other CoOOH facets, including (10-10). The calculations are carried out using a very basic computational approach, similar to that employed in earlier computational studies of the OER on CoOOH (e.g., ref. 59). Most notably, the authors explored a single OER mechanism, assumed the surface to be defect-free and ignored solvent effects. Remarkably, they do not even specify the DFT functional used for their calculations. It is fair to say that the reported calculations are below the current standards for computational studies of the OER on materials and the results obtained are both expected and inconclusive. In particular, I think that the authors need to explore the influence of defects on the OER activity. The key role of defects on the basal (0001) facet has been reported in recent computational studies of the OER on nickel oxyhydroxide, a material with characteristics similar to CoOOH (see, e.g., ref 39; another recent study is ACS Catal. 2022, 12, 295). Most importantly, the authors have actually observed defects on their CoOOH NSs; as reported at p. 7: "The slightly lower intensity of NS over NR coincides with the lower CN (5.5) for the Co-O path in CoOOH NS and should be attributed to the defects on the basal (0001) facets that dominate the surface of NS." With such experimental evidence, it seems obvious that a detailed exploration of the role of defects is essential in order to identify the active sites of the OER activity.

Reviewer #2:

Remarks to the Author:

The present study addresses the correlation between functional performance and the exposed facets of a model CoOOH OER electrocatalyst, synthesized as nanosheets (NS) and nanorods (NR). The authors have carried out a systematic structural, surface and functional study to understand that the catalytic active facet in this material is 10-10 and their findings are relevant to pave the way towards the rational design and surface engineering of electrocatalysts. However, there are some issues, which need further clarification before the manuscript can be accepted for publication.

- 1) Definitions of NR and NS must be done the first time they appear in the text. This is only done in the abstract, but it should be also done in the main text of the manuscript.
- 2) Identically, there is no definition of the symbols NS-S, NS-ML, NS-M and NS-L before Figure 1 is cited in the text.
- 3) Line 132 "...spanning over one order of magnitude..."
- 4) When discussing the EXAFS spectra of Figure XX, the authors claim that "...the lower intensity of NS over NR coincides with the lower coordination number (5,5) for the Co-O path in CoOOH NS and should be attributed to the defects of the basal (0001) facets that dominate the surface of NS." Surface defects are claimed as active catalytic sites in many electrocatalysts, and the authors should discuss the role of these defects on the functional performance of the studied materials.
- 5) Figure 4a does not show the activity relationship described in the main text. In particular the MS-ML and MS-L samples seem to be wrong in the Figure, if the ranking of overpotentials given in line 243 is correct. In the same line, the currents included in Figure 4b are not extracted from Figure 4a. At 450 mV overpotential, the current
- 6) The authors should also discuss their Tafel analysis with respect to relevant studies claiming the difficulties of this analysis in multielectron charge transfer processes, as OER. See for example: <https://doi.org/10.1038/srep13801>. Additionally, intrinsic electrocatalytic activity can be evaluated through impedance spectroscopy, as shown in <https://doi.org/10.1021/acs.jpcclett.9b01232> and the authors should comment their findings in relation with this and similar studies reported.

- 7) The authors should explain in more detail the implications of the power-law exponential dependence of catalytic current and geometrical dimension (d) of the NS and NR, since the extracted conclusions are not straightforward.
- 8) Line 314. There is a reference to Supplementary Figure 12, which is not correct, the authors refer to Supplementary Figure 10.
- 9) On the other hand, the message from Supplementary Figure 12 is not clear. There is not enough information in the main text and SI to understand the meaning and significance of this figure.
- 10) In Supplementary table S1, the authors provide the average sizes of the basal plane and the lateral plane from the analysis of 100 particles. Error intervals could be included in these statistical data to fully understand the statistical significance of the data.
- 11) The authors should demonstrate that the observed currents correspond to oxygen evolution through gas chromatography experiments.
- 12) All potentials in the manuscript and Supplementary Information should be reported versus RHE.

Response to referees

We thank the three referees for taking the time to carefully review the manuscript and for giving many useful suggestions. The manuscript has been revised according to their comments; the changes have been highlighted in the revised version. We feel the quality of the paper is much-improved thanks to the input from the referees. Below is a point-by-point response.

Referee #1

General comments: This manuscript reports an interesting experimental study of the OER on CoOOH nanosheets (NSs) and nanorods (NRs). The results show a higher activity on the NRs, suggesting a higher reactivity of the CoOOH (10-10) facet in comparison to the (0001) basal plane. To support their experimental results, the authors performed DFT calculations of the OER energetics, which show a higher overpotential on the (0001) surface than on other CoOOH facets, including (10-10).

Our response: We thank the referee for the positive feedback. We have conducted additional experiments and revised the paper according to the referee's suggestions (See below).

Comment 1: The calculations are carried out using a very basic computational approach, similar to that employed in earlier computational studies of the OER on CoOOH (e.g., ref. 59). Most notably, the authors explored a single OER mechanism, assumed the surface to be defect-free and ignored solvent effects. Remarkably, they do not even specify the DFT functional used for their calculations. It is fair to say that the reported calculations are below the current standards for computational studies of the OER on materials and the results obtained are both expected and inconclusive. In particular, I think that the authors need to explore the influence of defects on the OER activity. The key role of defects on the basal (0001) facet has been reported in recent computational studies of the OER on nickel oxyhydroxide, a material with characteristics similar to CoOOH (see, e.g., ref 39; another recent study is ACS. Catal. 2022, 12, 295). Most importantly, the authors have actually observed defects on their CoOOH NSs; as reported at p. 7: "The slightly lower intensity of NS over NR coincides with the lower CN (5.5) for the Co-O path in CoOOH NS and should be attributed to the defects on the basal (0001) facets that dominate the surface of NS." With such experimental evidence, it seems obvious that a detailed exploration of the role of defects is essential in order to identify the active sites of the OER activity.

Our response: Thanks for the good suggestion. We have updated our computational approach from a basic DFT to the Hubbard-corrected DFT (with solvation and D3 correction). It is now based on the Perdew–Burke–Ernzerhof (PBE) exchange–correlation (XC) functional (PBE+U)¹. Van der Waals (vdW) interactions are accounted for at the D3 level. The effective Hubbard-U parameter (U-J) of 3.47 eV is obtained by recovering the experimental oxidation energy for the reaction $6\text{CoO} + \text{O}_2 \rightarrow 2\text{Co}_3\text{O}_4$ ². The effect of incorporating water solvation is included in an implicit manner. It is performed by

making use of a dielectric polarizable continuum model (PCM), as implemented in the VASPsol package. A dielectric constant of 78.4 is used to simulate the solvation provided by water. The DFT functional has been mentioned in the revised manuscript now and the details about the theoretical calculation are shown in the Supplementary information. Both perfect and imperfect facets (with oxygen vacancies) are calculated. Besides the thermodynamic overpotentials, the coordination environments and electronic structures are investigated, in order to unveil the underlying relationship between them. The results are updated in Figure 5 and Supplementary Fig. 19-Fig. 28 of the revised manuscript. The discussion has been rewritten in the “Theoretical calculation of the geometrical active sites of CoOOH” part, with the inclusion of the influence of oxygen vacancies on the reactivities of different facets. The main results are shown below.

We first scrutinized the difference in the coordination of surface cobalt and oxygen atoms on basal (0001) facets and lateral (10 $\bar{1}$ 0) and (1 $\bar{1}$ 00) facets (Fig. 5a, left panel), followed by investigating their influence on electronic structures (Fig. 5a, right panel, Fig. 5b-5d). Basal (0001) facets are terminated by fully coordinated sites (CoFCS); Lateral (10 $\bar{1}$ 0) facets have only coordinatively unsaturated sites (CoCUS); Lateral (1 $\bar{1}$ 00) facets contain both CoFCS and CoCUS (Fig. 5a, left panel). Besides, the bonded oxygens show different coordination environments, varying from threefold coordinated oxygens O_{3c} to a mixture of twofold-coordinated oxygen atoms (O_{2c}) and O_{3c} (Fig. 5a, left panel, see more details in the “Theoretical calculation of the geometrical active sites of CoOOH” part of the revised manuscript). Theoretical calculations show that owing to the different coordination environments the lateral facets upshift the O 2p-band center closer to the Fermi level over that of the basal facets, accompanied by a reduced energy gap between the Co 3d and O 2p-band centers and a higher d-band center of cobalt (Fig. 5b and Supplementary Fig. 20). The changes in electronic structures indicate an enhanced binding energetics of Co sites with adsorbates and thus the greater covalency of the Co-O bond on lateral surface^{3,4}. To support this, the 2D electron localization function (ELF) and the crystal orbital Hamilton population (COHP) analysis were performed. The surface-adsorbed Co-O bond of lateral facets shows much less electron localization function (ELF) values (0.52 and 0.54) than that of basal facets (0.60), which suggests it is less polarized and exhibits less ionic feature (Fig. 5a, right panel)⁵. The integrals of COHP up to the Fermi level for lateral facets (3.70 and 3.71) are higher than that for basal facets (1.87), verifying the stronger Co-O bond on lateral facets (Fig. 5c)⁶. This is further evidenced by the short Co-O bond length and the higher bond dissociation energy calculated for lateral facets (Supplementary Table 5). The different hybridization configuration associated with a stronger metal-oxygen bond (Fig. 5d) indicates a higher reactivity of oxygen atoms and favorable catalytic kinetics for oxygen evolution reaction^{3,7}.

The DFT calculations were performed to investigate the thermodynamic overpotentials, which were then linked to the calculated electronic structures, to gain in-depth insight into the catalytic

mechanism of CoOOH nanocrystals at the PBE+U+D3 level. In agreement with experimental results, basal (0001) facets are inert, and lateral (10 $\bar{1}$ 0) facets are active for OER catalysis, corresponding to thermodynamic overpotentials of 1.95 V and 0.66 V, respectively (Fig. 5f-5i). The potential-determining step (PDS) shifts from desorption*O for basal facets to absorption of *O for lateral facets. This corresponds well with the above analysis of the electronic structures, which shows upshifting of O 2p states band center for fast OER catalysis of lateral facets^{3,7,8}. To further understand the facet's dependent activities, we studied the oxidation states of oxygen atoms and cobalt sites by analyzing the Bader charges and local magnetic moments (Fig. 5h-5j and Supplementary Table 6). The cobalt atoms transit to higher valence (Co⁺⁴ or Co⁺³⁻⁴) on lateral facets over that (Co⁺³) on basal facets in the OER catalysis. It results in a weakened absorption of reaction intermediates, in line with the PDS shift from desorption of *O for basal facets to absorption of *O for lateral facets. Additionally, DFT calculations were also performed without the inclusion of +U functional or implicit solvation. They show the same activity trend despite some differences in the absolute values (Supplementary Fig. 27 and Supplementary Table 5).

According to the reviewer's suggestion, we discussed the influence of oxygen vacancies on the reactivity of basal and lateral facets, given their presence in both CoOOH NR and NS. To support the theoretical calculation, the presence of oxygen vacancies has been confirmed by electron energy loss spectra (EELS) spectrum of oxygen K-edge and Co L-edge of COOH NS and NR (see Fig. 2d below, more details in the "Microstructures of CoOOH NR and NS" part of the revised manuscript). The XAS data have also been re-fitted more accurately to extract information about oxygen vacancies (see Supplementary Fig. 8 below, more details in the "Microstructures of CoOOH NR and NS" part of the revised manuscript). They suggest that a small portion of oxygen vacancies are present on basal and lateral facets, and the basal ones contain more. For theoretical calculations, oxygen vacancies that are nearest and second nearest to active Co sites (denoted as V_{O1} and V_{O2}, respectively, Fig. 5e) were investigated. The reaction paths are altered substantially in the presence of oxygen vacancies, where the thermodynamic overpotentials and PDS are dependent on the location of oxygen vacancies (Fig. 5f and 5g and Supplementary Fig. 21-26). On basal facets, oxygen vacancies result in a decrease of 0.41 V in the thermodynamic overpotential. The PDS retains at the desorption of *O (step 3), and the thermodynamic overpotential (1.54 V) is still much larger than that of lateral facets (0.66 V). This suggests the inertness of basal facets in the presence of oxygen vacancies. Several groups investigated the theoretical possibility of activating the basal facets of NiOOH (similar to CoOOH) using different models^{1,9}. Oxygen vacancies, of either V_{O1} or V_{O2}, have a minor influence on the reactivity of lateral (10 $\bar{1}$ 0) facets. A minimum overpotential of 0.61 V is observed, which is only 0.05 V lower than that for perfect (10 $\bar{1}$ 0) facets (0.66 V). Given the similar reactivity on perfect and imperfect lateral (10 $\bar{1}$ 0) facets, we suggest that the OER of CoOOH NR and NS proceeds with two mechanisms (involving oxygen vacancies or not). Additionally, we found that oxygen vacancies can substantially activate the

lateral (1100) facets (although the facets were not observed in our CoOOH NS and NR), yielding a thermodynamic overpotential of only 0.38 V. The introduction of oxygen vacancies results in an absorption configuration of *OOH with a Co-O-O three-membered ring and greatly lowers the formation energy of *OH by introducing a hydrogen bond with a neighboring hydroxyl group. This finding has an implication for the future structural engineering of OER catalysts; for instance, it indicates the potential of combining defect engineering and facet regulation to optimize the electronic structure for fast OER kinetics.

Figure 5 | Theoretical calculation of the active sites of crystalline CoOOH. (a) Schematics depicting the arrangement of surface atoms on (0001), (10 $\bar{1}$ 0), and (1 $\bar{1}$ 00) facets and their corresponding electron localization function (ELF) contour slices of the surface Co-O bond; (b) The electron density of states for the oxygen 2p orbitals of O_{2c}/O_{3c} atoms (shaded) and O_{1c} atoms. (c) Crystal orbital Hamilton populations (COHP) of the surface Co-O bond (CocUS-O_{1c} for lateral facets and CocUS-O_{3c} for basal facets). (d) Schematic band diagrams of the basal and lateral facets. (e) Schematics of investigated facets and oxygen vacancies on them. (f) The free-energy landscape of the OER on different facets of CoOOH *R3m* crystal (at the PBE+U+D3 level of theory). (g) 2D map of

theoretical overpotentials η . Cubes are for perfect facets. Triangles and circles are for facets with V_{O1} and V_{O2} , respectively. (h, i, j) The proposed OER proceeding on (h) (0001), (i) (10 $\bar{1}$ 0), and (j) (1 $\bar{1}$ 00) facets.

Lastly, the calculated adsorption energies for both perfect and imperfect surfaces are plotted as scaling relations between ΔG_{OOH} and ΔG_{OH} , to exhibit a general picture of the facets' dependent activity and the relation to the electronic structures (Supplementary Fig. 28a). It gives a relationship of $\Delta G_{OOH} = 0.69\Delta G_{OH} + 3.20$ eV ($R^2 = 0.95$), corresponding well with reported scaling plots of transition metals oxides^{10, 11}. Basal and lateral facets (no oxygen vacancy) are located on the left and right legs of the established volcano plot, respectively (Supplementary Fig. S28b). They are more than 0.28 V away from the volcano's top site. The introduction of oxygen vacancies further regulates the reactivity under the traditional scheme for OER catalysis, capable of pushing the active sites closer to the volcano top site as suggested by the imperfect lateral (1 $\bar{1}$ 00) facets in our calculation (Supplementary Fig. S28b).

Figure 2| X-ray absorption, electron energy loss, and X-ray photoemission spectra of CoOOH NS and NR. (d) EELS spectrum of oxygen K-edge and Co L-edge of COOH NS and NR.

Supplementary Figure 8 | (a) R space and (b) K space curve fitting on CoOOH NS; (c) R space and (d) K space curve fitting on CoOOH NR.

Referee #2

General comments: The present study addresses the correlation between functional performance and the exposed facets of a model CoOOH OER electrocatalyst, synthesized as nanosheets (NS) and nanorods (NR). The authors have carried out a systematic structural, surface, and functional study to understand that the catalytic active facet in this material is 10-10 and their findings are relevant to pave the way towards the rational design and surface engineering of electrocatalysts. However, there are some issues, which need further clarification before the manuscript can be accepted for publication.

Our response: We thank the referee for the positive feedback. We have conducted additional experiments and revised the paper according to the referee's suggestions (See below).

Comment 1: Definitions of NR and NS must be done the first time they appear in the text. This is only done in the abstract, but it should be also done in the main text of the manuscript.

Our response: Thanks for the suggestion. We have defined NR (nanosheets) and NS (nanorods) in the last paragraph of the "Introduction" part, where they first appear in the main text of the revised manuscript.

Definitions of NR and NS in the last paragraph of the "Introduction" part:

"To bypass the dynamic surface reconstruction, we chose crystalline CoOOH as the model catalyst, because they were previously demonstrated to be catalytically and structurally stable. The size- and morphology-dependent catalytic activities suggested a ~50 folds difference in specific activities between nanorods (NR) and nanosheets (NS)."

Comment 2: Identically, there is no definition of the symbols NS-S, NS-ML, NS-M and NS-L before Figure 1 is cited in the text.

Our response: Thanks for the suggestion. We have defined NS-S, NS-ML, NS-M, and NS-L in the "Synthesis of CoOOH NR and NS" part of the revised manuscript. The definitions are now present at their first appearance.

Definitions of NS-S, NS-ML, NS-M, and NS-L in the "Synthesis of CoOOH NR and NS" part:

"For size-dependent and correlation analysis, four lateral sizes ranging over one order of magnitude were synthesized for CoOOH NS, denoting NS-S (small nanosheets: $0.10 \pm 0.04 \mu\text{m}$), NS-M (medium nanosheets: $0.5 \pm 0.08 \mu\text{m}$), NS-ML (medium-large nanosheets: $1.2 \pm 0.3 \mu\text{m}$), and NS-L (large nanosheets: $3.7 \pm 1.4 \mu\text{m}$). The particle sizes were averaged from measurements over around 100 particles (Supplementary Table 1)."

Comment 3: Line 132 "...spanning over one order of magnitude...".

Our response: The sentence has been changed to "ranging from ~100 nm to ~5 μm ". In addition, we

have polished the manuscript to eliminate grammar errors and typos.

Comment 4: When discussing the EXAFS spectra of Figure XX, the authors claim that “...the lower intensity of NS over NR coincides with the lower coordination number (5,5) for the Co-O path in CoOOH NS and should be attributed to the defects of the basal (0001) facets that dominate the surface of NS.” Surface defects are claimed as active catalytic sites in many electrocatalysts, and the authors should discuss the role of these defects on the functional performance of the studied materials.

Our response: According to the reviewer’s suggestion, we discussed the influence of oxygen vacancies on the reactivity of basal and lateral facets, given that they are present in both CoOOH NR and NS. To support the theoretical calculation, the presence of oxygen vacancies has been confirmed by electron energy loss spectra (EELS) spectrum of oxygen K-edge and Co L-edge of COOH NS and NR (see details in the “Microstructures of CoOOH NR and NS” part of the revised manuscript). The XAS data have also been re-fitted more accurately to extract information about oxygen vacancies (see details in the “Microstructures of CoOOH NR and NS” part of the revised manuscript). They suggest that a small portion of oxygen vacancies are present on basal and lateral facets, and the basal ones contain more. For theoretical calculations, oxygen vacancies that are nearest and second nearest to active Co sites (denoted as V_{O1} and V_{O2} , respectively, Fig. 5e) were investigated. The reaction paths are altered substantially in the presence of oxygen vacancies, where the thermodynamic overpotentials and PDS are dependent on the location of oxygen vacancies (Fig. 5f and 5g and Supplementary Fig. 21-26). On basal facets, oxygen vacancies result in a decrease of 0.41 V in the thermodynamic overpotential. The PDS retains at the step of desorption of *O (step 3), and the thermodynamic overpotential (1.54 V) is still much larger than that of lateral facets (0.66 V). It suggests the inertness of basal facets even though oxygen vacancies are accounted for. Several groups investigated the theoretical possibility of activating the basal facets of NiOOH (similar to CoOOH) using different models^{1,9}. Oxygen vacancies, of either V_{O1} or V_{O2} , have a minor influence on the reactivity of lateral (10 $\bar{1}0$) facets. A minimum overpotential of 0.61 V is observed, which is only 0.05 V lower than that for perfect (10 $\bar{1}0$) facets (0.66 V). Given the similar reactivity on perfect and imperfect lateral (10 $\bar{1}0$) facets, we suggest that the OER of CoOOH NR and NS proceeds with two mechanisms (involving oxygen vacancies or not). Additionally, we found that oxygen vacancies can substantially activate the lateral (1 $\bar{1}00$) facets (although the facets were not observed in our CoOOH NS and NR), yielding a thermodynamic overpotential of only 0.38 V. The introduction of oxygen vacancies results in an absorption configuration of *OOH with a Co-O-O three-membered ring and greatly lowers the formation energy of *OH by introducing a hydrogen bond with a neighboring hydroxyl group. This finding has a implication for the structural engineering of OER catalysts; for instance, it indicates the potential of combining defect engineering and facet regulation to optimize the electronic structure for fast OER kinetics. The data has been updated in Figure 5 and Supplementary Fig. 21-26 of the revised manuscript (see also in the Response to Comment 1 from referee 1 above). The discussion has been added to the “Theoretical calculation of the geometrical active sites of CoOOH” part of the revised manuscript.

Comment 5: Figure 4a does not show the activity relationship described in the main text. In particular the NS-ML and NS-L samples seem to be wrong in the Figure, if the ranking of overpotentials given in line 243 is correct. In the same line, the currents included in Figure 4b are not extracted from Figure 4a. At 450 mV overpotential, the current.

Our response: We thank the referee for the careful review and for pointing out the mistakes. The polarization curves of NS-ML and NS-L were labeled in the wrong order in Fig. 4a of the previous manuscript. We have corrected the mistake in the revised one (see also below). The current densities in Fig. 4b were extracted from Fig. 4a at the overpotential of 425 mV rather than 450 mV. We have corrected it in the revised manuscript (see also below).

Figure 4 | Catalytic activities of CoOOH NR and CoOOH NS for oxygen evolution reaction. (a) Polarization curves. The inset shows the corresponding Tafel plots; (b) Size-dependent activities of NS at the same loadings;

Comment 6: The authors should also discuss their Tafel analysis with respect to relevant studies claiming the difficulties of this analysis in multielectron charge transfer processes, as OER. See for example: <https://doi.org/10.1038/srep13801>.

Our response: We thank the referee for the suggestion. As the reviewer mentioned, the kinetic interpretation of the Tafel slope is controversial to date, in particular for reactions coupled with multiple electron transfers¹². Given that our materials were synthesized from the same kind of precursory materials ($\text{Co}(\text{OH})_2$) and via the same topochemical oxidation process, it is rather fair to assume that the products have similar surface electronic structures and thus proceed with the OER via the same catalytic mechanism. In addition, as the current densities were recorded at relatively lower overpotentials (corresponding to low current densities), the coverage of the adsorbed species should be low, thereby allowing the Tafel analysis in conjunction with the Butler-Volmer equation¹². A Tafel slope of 60 mV, usually associated with a catalytic mechanism involving chemical reaction as the rate-

determining step after the first electron transfer¹³, was experimentally observed for all the CoOOH samples, regardless of their sizes and morphologies. The size-independent mechanism and the low coverage of the adsorbed species, therefore, guarantee a quantitative correlation analysis of catalytic activities with geometric surface areas. The discussion of the Tafel slope has been added to the revised manuscript (see also below).

Tafel analysis in the “Correlating catalytic activities with structure parameters” part:

“The Tafel slope is a kinetic parameter, which has an implication on the catalytic mechanism. Though the interpretation is controversial (in particular for reaction coupled with multiple electron transfer), a Tafel slope of 60 mV could be associated with a catalytic mechanism involving chemical reaction as the rate-determining step after the first electron transfer¹⁶. Given the same kind of precursory materials (Co(OH)₂) and the same topochemical oxidation process for materials synthesis, it is rather fair to assume the samples possess similar surface electronic structures and states and thus proceed with the OER via the same catalytic mechanism. In addition, the current densities were recorded at a relatively lower overpotential, which suggests the coverage of the adsorbed species should be low, thereby allowing the Tafel analysis in conjunction with the Butler-Volmer equation⁵⁰. The size-independent mechanism and the low coverage of the adsorbed species guarantee a quantitative correlation analysis of catalytic activities with geometric surface areas.”

Comment 7: Additionally, intrinsic electrocatalytic activity can be evaluated through impedance spectroscopy, as shown in <https://doi.org/10.1021/acs.jpcclett.9b01232> and the authors should comment on their findings in relation to this and similar studies reported.

Our response: According to the referee’s suggestion, the turnover frequencies (TOFs) of CoOOH NR and NS have been evaluated through electrochemical impedance spectroscopy (EIS) following the method in the suggested paper¹⁴. The CoOOH NR and NS show a similar TOF_{EIS} at a set of applied potentials (Supplementary Fig. S17d), confirming the intrinsic reactivity of the lateral facets. The discussion has been added to the revised manuscript (see also below). The data are added to Supplementary Fig. S17 in the revised manuscript (see also below). The details about the characterization are added to the “Method-1.4 Electrochemical test” part in the Supplementary information.

Discussion of TOF_{EIS} and its relation with our findings in the revised manuscript:

“The reactivity of lateral facets is further confirmed by their similar TOF_{EIS} extracted from the EIS, which is a measure of the intrinsic activity (Supplementary Fig. 13 and Supplementary Table 3)⁵¹.”

Supplementary Figure 13 | Electrochemical impedance spectroscopy analysis of CoOOH NR and CoOOH NS. (a-b) The Nyquist plots for the impedance response of (a) NS-ML and (b) NR. (c) Plot of $\log(R_{CT}^{-1})$ vs. potentials, showing Tafel slopes of c.a. 60 mV/dec. (d) TOF_{EIS} of NS-ML and NR at a set of applied potentials. The inset in (d) shows the equivalent circuit model for EIS fitting. R_s stands for series resistance; R_T stands for catalyst's charge-transport resistance; R_{CT} stands for the charge-transfer resistance at the catalyst/solution interface; C_μ stands for the chemical capacitance.

Comment 8: The authors should explain in more detail the implications of the power-law exponential dependence of catalytic current and geometrical dimension (d) of the NS and NR, since the extracted conclusions are not straightforward.

Our response: We thank the referee for the suggestion. We have rewritten the discussion about the power-law exponential dependence of catalytic current on geometrical dimension. The equation for calculating the catalytic activities (current densities at a specific overpotential) is provided in the revised manuscript (equation 1, see also below), to clarify the structural model and the relation to activities. The discussion on the correlation (the power-law exponential dependence) is updated in the revised manuscript (see also below).

The overall activities of catalysts are the sum of activities of all the exposed facets (they include

basal facets and lateral facets for CoOOH NR and NS). It can be described by equation 1 below.

$$J_{\text{overall}} = J_b + J_l = j_b \times S_b + j_l \times S_l = \frac{j_b m}{\rho h} + \frac{8j_l m}{\sqrt{3}\rho d} \quad 1)$$

where J_{overall} , J_b , and J_l are the overall, basal, and lateral current, respectively; j_b and j_l are the specific current densities on basal and lateral planes, respectively; S_b and S_l are the basal and lateral surface area, respectively; m and ρ are the mass and density of catalysts, respectively; h and d are the height and basal size of NS, respectively. The deduction of equation 1 has been included in the “*Note 5: Deduction of power-law exponent relationship*” part of the Supplementary information. This equation indicates that the current density is ideally proportional to the reciprocal basal size $1/d$, given other parameters including j_b , j_l and h are the same for all the CoOOH NS. The experimental correlation was discussed in conjunction with equation 1. It follows a power-law exponent relationship ($J = J_0 + k \cdot d^{-n}$, see discussion in Supplementary Note 5) and gives a fitting power of around 0.5. Ideally, the correlation indicates the geometric location of catalytic active sites (Fig. 4b): for $n = 1$ (activities are proportional to the reciprocal basal sizes), active sites are located only on lateral facets if $J_0 = 0$ (corresponding to $j_b=0$ and $j_l \neq 0$ in equation 1) and on both lateral and basal facets if $J_0 \neq 0$ (corresponding to $j_b \neq 0$ and $j_l \neq 0$ in equation 1); for $n = 0$ (activities are independent on the sizes of basal facets), active sites are on basal facets only (corresponding to $j_b \neq 0$ and $j_l = 0$ in equation 1). The fitting power of $n = 0.5$ suggests the reactivity of lateral facets but cannot tell whether the basal facets are active or not. The deviation of fitting power could be attributed to the particle agglomeration, which is inevitable at mass loadings of dozens of micrograms and results in electrically unavailable regions. To further test the reactivity of the basal surface, the morphology-dependent activities were studied on NR and NS (see more details in the “Correlating catalytic activities with structure parameters” part of the revised manuscript).

Scheme S1. Structural models of (a) CoOOH NS and (b) CoOOH NR.

Figure 4 | Catalytic activities of CoOOH NR and CoOOH NS for oxygen evolution reaction. (b) Size-dependent activities of NS at the same loadings;

Discussion on the correlation (the power-law exponential dependence) in the “Correlating catalytic activities with structure parameters” part:

“The current densities of NR and NS are dependent on the intrinsic activity and the surface area of each specific plane (basal and lateral plane here). They are calculated from equation 1,

$$J_{overall} = J_b + J_l = j_b \times S_b + j_l \times S_l = \frac{j_b m}{\rho h} + \frac{8j_l m}{\sqrt{3}\rho d} \quad (1)$$

where $J_{overall}$, J_b , and J_l are the overall, basal, and lateral current density; j_b and j_l are the specific current densities on basal and lateral planes; S_b and S_l are the basal and lateral surface area; m and ρ are the mass and density of catalysts, respectively; h and d are the height and basal size of NS (detailed discussion is shown in Supplementary Note 4 and Note 5). Given the same values of j_b , j_l , m , ρ , and h for all the samples, the equation indicates that their overall current densities are proportional to the reciprocal basal size ($1/d$). Fig. 4b shows the experimental correlation of current densities with the reciprocal basal sizes. It follows a power-law exponent relationship ($J = J_0 + k \cdot d^{-n}$, see discussion in Supplementary Note 5) and gives a fitting power of around 0.5. Ideally, the correlation could indicate the geometric location of catalytic active sites (Fig. 4b and Supplementary Note 5): for $n = 1$ (activities are proportional to the reciprocal basal sizes), active sites are located only on lateral facets if $J_0 = 0$ (corresponding to $j_b=0$ and $j_l \neq 0$ in equation 1) and on both lateral and basal facets if $J_0 \neq 0$ (corresponding to $j_b \neq 0$ and $j_l \neq 0$ in equation 1); for $n = 0$ (activities are independent on the sizes of basal facets), active sites are on basal facets only (corresponding to $j_b \neq 0$ and $j_l = 0$ in equation 1). The fitting power of $n = 0.5$ indicates the reactivity of lateral facets but cannot tell whether the basal facets are active or not. The deviation of fitting power could be attributed to the particle agglomeration, which is inevitable at mass loadings of dozens of micrograms and results in electrically unavailable regions.”

Comment 9: Line 314. There is a reference to Supplementary Figure 12, which is not correct, the authors refer to Supplementary Figure 10.

Our response: Thanks for the careful reading and pointing out the mistake. We have corrected the citation in the revised manuscript.

Revision of the reference to Supplementary Figure 12:

“The polarization curves were recorded at 2-3 loadings for each kind of sample (Supplementary Fig. 15). The current densities at an overpotential of 450 mV were used to represent the activities.”

Comment 10: On the other hand, the message from Supplementary Figure 12 is not clear. There is not enough information in the main text and SI to understand the meaning and significance of this figure.

Our response: Thanks for the suggestion. For clarity, we add an introduction to Supplementary Figure 12 (corresponding to Supplementary Figure 15 in the revised manuscript). It shows the polarization curves of CoOOH NS and NR at 2-3 loadings for each kind. The current densities at an overpotential of 450 mV were used to represent the activities to correlate with their overall, lateral, and basal surface areas. Impressively, the activities of all the samples, regardless of the sizes and shapes, show a strong correlation with the lateral facet areas, rather than the basal surface areas. This firmly supports the reactivity of lateral facets and the inertness of basal facets. The discussion is added to the revised manuscript (see also below).

Description of Supplementary Figure 12 and the discussion by linking it to correlation analysis:

“The polarization curves were recorded at 2-3 loadings for each kind of sample (Supplementary Fig. 15). The current densities at an overpotential of 450 mV were used to represent the activities. They were first correlated with their overall surface areas. For each sample, the activities increase linearly with surface areas (Fig. 4c, 4d, and Supplementary Fig. 16). The linear relation suggests that the electrochemical analysis eliminated the interference from mass and charge transport at submonolayer loadings owing to the discrete dispersion of nanoparticles. It is worth noting that CoOOH NR exhibits a linear slope of 3.7-6.4 times that of NS and the linear slopes of NS are size-dependent (Fig. 4c and Supplementary Fig. 16a and 16b). The morphology- and size-dependent linear slopes confirm the facet-dependent activity, that is, lateral facets (dominating in CoOOH NR) are much more active than basal facets (dominating in CoOOH in NS). To elucidate the role of each specific facet, we correlated the current densities with lateral and basal surface area, respectively. Impressively, the activities of all the samples, regardless of the sizes and shapes, show a strong correlation with the lateral facet areas, rather than the basal surface areas. (Fig. 4d and Supplementary Fig. 16c). The perfect linear fitting strongly supports the reactivity of lateral facets and the inertness of basal facets.”

Comment 11: In Supplementary table S1, the authors provide the average sizes of the basal plane and the lateral plane from the analysis of over 100 particles. Error intervals could be included in these statistical data to fully understand the statistical significance of the data.

Our response: We thank the referee for the suggestion. We have included error intervals in Supplementary Table 1 in the revised manuscript (see also below).

Supplementary Table 1. The statistical data of particle sizes and roughness of CoOOH Samples. The particle sizes were based on measurements over an average of >100 particles.

Sample Name	Average Basal Plane Size (μm)	Average	
		Thickness/Length of Lateral Plane (μm)	Roughness (nm)
CoOOH NS-L	3.76 ± 1.42	0.056 ± 0.08	0.62
CoOOH NS-ML	1.19 ± 0.37	0.062 ± 0.04	0.82
CoOOH NS-M	0.49 ± 0.08	0.063 ± 0.03	1.35
CoOOH NS-S	0.11 ± 0.04	0.058 ± 0.04	1.13
CoOOH NR	0.83 ± 0.24	5.825 ± 2.31	N.A.

Comment 12: The authors should demonstrate that the observed currents correspond to oxygen evolution through gas chromatography experiments.

Our response: We thank the referee for the suggestion. The faradaic efficiencies were tested by gas chromatography experiments. The average faradaic efficiencies over 1-hour electrolysis at $10 \text{ mA} \cdot \text{cm}^{-2}$ are 98% for NS and 95% for NR, comparable to an IrO_2 benchmark catalyst (Supplementary Fig. 10b and 10c). In addition, the drainage method and rotating ring-disc electrode (RRDE) methods are applied to confirm the faradaic efficiencies. In the drainage method, the evolved H_2 and O_2 exhibit a volume ratio of nearly 2:1, which follows closely with the theoretical value (Supplementary Fig. 10a). The RRDE method shows Faradaic efficiencies of >98% for NS and NR (Supplementary Fig. 10d-10i). The description has been added to the revised manuscript (see also below) and Supplementary Fig. 10 in the Supplementary Information.

Description of the faradaic efficiencies of NS and NR in the revised manuscript:

“Gas chromatography experiments, drainage method, and rotating ring-disc electrode (RRDE) method revealed Faradaic efficiencies of >95% for CoOOH NS and NR, suggesting that the anodic currents were exclusively for the OER catalysis (Supplementary Fig. 10).”

Supplementary Figure 10 | Measurement of the Faradaic efficiency. (a) Photographs of water splitting electrolyzer during the electrolysis (drainage method). (b) Gas chromatography curves of gaseous products from the OER catalyzed by NS-M, NR, and commercial IrO₂ catalyst loaded on carbon fiber paper. (c) Experimental and theoretical volumes of O₂ gases during the electrolysis. (d-h) Ring current of (d) NS-S, (e) NS-M, (f) NS-ML, (g) NS-L, (h) NR on an RRDE (1500 r. p.m.) in N₂-saturated 1 M KOH solution (ring potential: 0.38 V). (i) Calculated faradaic efficiency of NS and NR via RRDE method.

Comment 13: All potentials in the manuscript and Supplementary Information should be reported versus RHE.

Our response: We thank the referee for the suggestion. We have converted the potentials to the RHE scale in the revised manuscript (see also below).

In the main text

“Figure 3 | In situ Raman and EC-AFM analysis of CoOOH NS and NR. (a) Raman spectra under applied potentials ranging from 1.03 V to 1.73 V vs. RHE (from bottom to top, with the interval of 0.1 V); (b) AFM images and (c) corresponding height of NS under applied potentials ranging from 1.03 V

to 1.73 V vs. RHE. The corresponding CV curve was also shown in (c). AFM measurement was interfered with by the evolving bubbles at potentials higher than 1.53 V vs. RHE.”

In Supplementary Information

“The potential window of CV scans was 1.224~0.1.324 V vs. RHE.”

“The double-layer capacitance (Cdl) was estimated by plotting the current densities of $\Delta j = j_a - j_c$ at 1.274V vs. RHE against the scan rates.”

Reference

1. Hu, Q., et al. Structure and oxygen evolution activity of β -NiOOH: Where Are the Protons? *ACS Catal.* **12**, 295-304 (2022).
2. García-Mota, M., Bajdich, M., Viswanathan, V., Vojvodic, A., Bell, A. T. Nørskov, J.K. Importance of correlation in determining electrocatalytic oxygen evolution activity on cobalt oxides. *J. Phys. Chem. C* **116**, 21077-21082 (2012).
3. Grimaud, A., et al. Activating lattice oxygen redox reactions in metal oxides to catalyse oxygen evolution. *Nat. Chem.* **9**, 457-465 (2017).
4. Grimaud, A., et al. Double perovskites as a family of highly active catalysts for oxygen evolution in alkaline solution. *Nat. Commun.* **4**, (2013).
5. Becke, A. D. & Edgecombe, K. E. A simple measure of electron localization in atomic and molecular systems. *J. Chem. Phys.* **92**, 5397-5403 (1990).
6. He, Z., et al. Activating lattice oxygen in NiFe-based (oxy)hydroxide for water electrolysis. *Nat. Commun.* **13**, (2022).
7. Grimaud, A., Hong, W. T., Shao-Horn, Y. & Tarascon, J.-M. Anionic redox processes for electrochemical devices. *Nat. Mater.* **15**, 121-126 (2016).
8. Kim, M. et al. Promotion of electrochemical oxygen evolution reaction by chemical coupling of cobalt to molybdenum carbide. *Appl. Catal. B Environ.* **227**, 340-348 (2018).
9. Govind Rajan, A., Martirez, J.M.P, Carter, E.A. Facet-independent oxygen evolution activity of pure β -NiOOH: different chemistries leading to similar overpotentials. *J. Am. Chem. Soc.* **142**, 3600-3612 (2020).
10. Rossmeisl, J., Qu, Z.-W., Zhu, H., Kroes, G.-J. & Nørskov, J. K. Electrolysis of water on oxide surfaces. *J. Electroanal. Chem.* **607**, 83-89 (2007).
11. Calle-Vallejo, F., et al. Number of outer electrons as descriptor for adsorption processes on transition metals and their oxides. *Chem. Sci.* **4**, 1245-1249 (2013).
12. Shinagawa, T., Garcia-Esparza, A.T., Takanabe, K., Insight on Tafel slopes from a microkinetic analysis of aqueous electrocatalysis for energy conversion. *Sci. Rep.* **5**, (2015).
13. Moysiadou, A., Lee, S., Hsu, C.-S., Chen, H.M., Hu, X. Mechanism of oxygen evolution catalyzed by cobalt oxyhydroxide: cobalt superoxide species as a key intermediate and dioxygen release as a rate-determining step. *J. Am. Chem. Soc.* **142**, 11901-11914 (2020).
14. Singh, C., Liberman, I., Shimoni, R., Ifraemov, R., Hod, I. Pristine versus pyrolyzed metal-organic

framework-based oxygen evolution electrocatalysts: evaluation of intrinsic activity using electrochemical impedance spectroscopy. *J. Phys. Chem. Lett.* **10**, 3630-3636 (2019).

Reviewers' Comments:

Reviewer #1:

Remarks to the Author:

I think the authors have adequately addressed my previous comments and concerns. The revised manuscript includes many new experimental and computational results and its quality has improved substantially.

Reviewer #2:

Remarks to the Author:

The authors have convincingly addressed all the issues raised by this referee. The manuscript can be accepted for publication.

Response to referees

We thank the three referees for taking the time to carefully review the manuscript and for giving positive comments. Below is a point-by-point response.

Referee #1

General comments: I think the authors have adequately addressed my previous comments and concerns. The revised manuscript includes many new experimental and computational results and its quality has improved substantially.

Our response: We thank the referee for the positive feedback.

Referee #2

General comments: The authors have convincingly addressed all the issues raised by this referee. The manuscript can be accepted for publication.

Our response: We thank the referee for the positive feedback.